Methods

# A high-content endogenous GLUT4 trafficking assay reveals new aspects of adipocyte biology

Alexis Diaz-Vegas[1],*, Dougall M Norris[2],*, Sigrid Jall-Rogg[1], Kristen C Cooke[1], Olivia J Conway[2], Amber S Shun-Shion[2], Xiaowen Duan[2], Meg Potter[1], Julian van Gerwen[1], Harry JM Baird[2], Sean J Humphrey[1], David E James[1,3], Daniel J Fazakerley[2], James G Burchfield[1]

**Insulin-induced GLUT4 translocation to the plasma membrane in muscle and adipocytes is crucial for whole-body glucose homeostasis. Currently, GLUT4 trafficking assays rely on overexpression of tagged GLUT4. Here we describe a high-content imaging platform for studying endogenous GLUT4 translocation in intact adipocytes. This method enables high fidelity analysis of GLUT4 responses to specific perturbations, multiplexing of other trafficking proteins and other features including lipid droplet morphology. Using this multiplexed approach we showed that Vps45 and Rab14 are selective regulators of GLUT4, but *Trarg1*, *Stx6*, *Stx16*, *Tbc1d4* and *Rab10* knockdown affected both GLUT4 and TfR translocation. Thus, GLUT4 and TfR translocation machinery likely have some overlap upon insulin-stimulation. In addition, we identified Kif13A, a Rab10 binding molecular motor, as a novel regulator of GLUT4 traffic. Finally, comparison of endogenous to overexpressed GLUT4 highlights that the endogenous GLUT4 methodology has an enhanced sensitivity to genetic perturbations and emphasises the advantage of studying endogenous protein trafficking for drug discovery and genetic analysis of insulin action in relevant cell types.**

## Introduction

The protein composition of the plasma membrane (PM) is dynamically regulated to maintain both cellular and organismal homeostasis. Transmembrane protein localisation to the cell surface is determined by the secretory and endocytic recycling pathways. A subset of protein cargoes trafficked through these pathways are subject to additional regulation by external cues and signal transduction pathways to control their abundance at the PM. Insulin-stimulated trafficking of the glucose transporter GLUT4 is a prime example of how signal transduction and the endomembrane system coordinate appropriate subcellular localization of specific proteins with the needs of an organism. GLUT4 continuously recycles through several intracellular compartments such as the TGN and endosomes, but in the absence of insulin or at low insulin concentrations, GLUT4 accumulates in intracellular sequestration vesicles known as GLUT4 storage vesicles (GSVs). Insulin signalling targets these GSVs, mobilising GLUT4 to the PM (Klip et al, 2019) and promoting glucose uptake by increasing the maximal glucose transport rate across the PM. This process is critical to glucose homeostasis (Stenbit et al, 1997), and is impaired in insulin resistance and type 2 diabetes (James et al, 2021). The rapid mobilisation of GLUT4 to the PM in response to insulin stimulation, and its relevance to metabolic disease, makes GLUT4 an ideal candidate to study regulated protein trafficking.

Studies of GLUT4 traffic have been hindered by technical limitations rooted in the inability to explore endogenous GLUT4 localization and translocation with sufficient temporal and spatial resolution, or in high-throughput for screening purposes. For example, biochemical techniques such as membrane subfractionation are not amenable to highly resolved temporal studies and lack spatial resolution due to impure fractions and the inability to discern between GLUT4 inserted versus not inserted in the PM (i.e., PM-associated vesicles). Microscopy-based approaches have provided important quantitative and spatial information regarding GLUT4 vesicles in both basal and insulin-stimulated states (Quon et al, 1994; Marsh et al, 1995; Rea & James, 1997; Malide et al, 2000; Martin et al, 2000). However, assessing endogenous GLUT4 localisation by electron microscopy is limited in throughput, and immunofluorescence microscopy using antibodies to intracellular epitopes on endogenous GLUT4 does not distinguish GLUT4 molecules embedded within the PM from those closely associated with the PM.

Fluorescent protein- and epitope-tagged GLUT4 reporter constructs have overcome many of these issues, allowing studies of GLUT4 traffic in live cells, at both the population and single-cell levels, with high spatiotemporal resolution. For example, GLUT4 constructs that contain an epitope tag within its first exofacial/

[1]Charles Perkins Centre, School of Life and Environmental Sciences, University of Sydney, Sydney, Australia   [2]Metabolic Research Laboratories, Wellcome-Medical Research Council Institute of Metabolic Science, University of Cambridge, Cambridge, UK   [3]School of Medical Sciences, University of Sydney, Sydney, Australia

Correspondence: james.burchfield@sydney.edu.au; djf72@medschl.cam.ac.uk; david.james@sydney.edu.au
Dougall M Norris's present address is School of Biotechnology and Biomolecular Sciences, The University of New South Wales, Sydney, Australia.
*Alexis Diaz-Vegas and Dougall M Norris contributed equally to this work.

lumenal loop allows specific detection of GLUT4 within the PM in non-permeabilized cells (Konrad et al, 2002; Govers et al, 2004; Karylowski et al, 2004). However, use of these GLUT4 fusion proteins requires caution because of potential overexpression artefacts or disruption of normal GLUT4 trafficking, depending on the position of the tag/epitope (Dobson et al, 1996; Al-Hasani et al, 1999; Prelich, 2012; Brewer et al, 2019). Indeed, GLUT4 overexpression protects against insulin resistance (Atkinson et al, 2013), affects stability of LNPEP/IRAP and VAMP2 (other proteins that reside in GSVs) (Carvalho et al, 2004). Despite these limitations to using GLUT4 fusion proteins, no new methods to study endogenous GLUT4 trafficking have been developed for more than 20 yr.

Recently, Tucker et al (2018) generated antibodies that recognize an extracellular epitope in endogenous GLUT4. Here, we explored the use of these reagents as a novel approach for assessing delivery of endogenous GLUT4 to the cell surface at high spatial resolution and in a high-throughput fashion. First, we established the use of these antibodies to characterize endogenous GLUT4 responses to insulin in cultured murine and human adipocytes. Second, we scaled this approach and demonstrated its applicability for drug screening, including developing new models of insulin resistance in cultured adipocytes. Third, we combined this approach with an optimized method for siRNA delivery in differentiated adipocytes to discover that the kinesin motor protein Kif13A regulates insulin-induced GLUT4 traffic. Finally, we established a high-content screening platform to simultaneously study two insulin-regulated proteins, GLUT4 and TfR as well as adipocyte lipid droplet content. Using this platform, we (1) identified specific and general regulators of GLUT4 and TfR trafficking and (2) determined that endogenous GLUT4 trafficking is more sensitive to knockdown of regulatory proteins than overexpressed HA-GLUT4-mRuby3. Overall, the analytical tools developed here will benefit the GLUT4 field, and are relevant to studies of protein trafficking using high content imaging approaches more generally.

## Results

### Characterizing endogenous GLUT4 translocation using antibodies targeting exofacial epitopes in GLUT4

Antibodies that label an outward facing epitope in GLUT4 and therefore label PM localised GLUT4 in non-permeabilized cells have long been sought for studying GLUT4. The absence of such tools led to use of overexpressed epitope- or fluorescent protein-tagged GLUT4 reporters (e.g., HA-GLUT4-GFP and Myc-GLUT4-GFP). Tucker et al (2018) recently developed antibodies that recognise exofacial epitopes in endogenous GLUT4 (Tucker et al, 2018). These antibodies provide a potentially powerful approach to detect and quantify transport-competent endogenous GLUT4 within the PM (LM052, LM059, and LM048) (Tucker et al, 2018). To investigate whether these antibodies could be used to quantify insulin-induced GLUT4 translocation, we used mouse 3T3-L1 adipocytes because these cells express high levels of GLUT4 and are highly insulin responsive. LM048 gave the strongest most consistent staining in 3T3-L1 adipocytes (Figs 1A and S1A), with 100 nM insulin invoking a fivefold increase in cell surface GLUT4, (Figs 1A and S1A).

siRNA-mediated knockdown of *Slc2a4*/*Glut4* in 3T3-L1 adipocytes reduced both total GLUT4 (using an antibody to the c-terminus of GLUT4) and LM048 staining, supporting the specificity of this approach (Fig 1B and C). In human SGBS adipocytes, LM052 gave the strongest staining of PM GLUT4, but the LM048 antibody was less effective (Fig S1B and C).

Insulin dose-dependently increased endogenous GLUT4 at the PM (EC50 = 0.860 nM), reaching its maximal effect at 10 nM insulin (Fig 1D and E). This is consistent with the effects of insulin on glucose transport and GLUT4 translocation measured by subcellular fractionation in the same cells (Piper et al, 1991). The EC50 for GLUT4 translocation was sensitive to assay buffers, as previously described (Shechter & Ron, 1986), with DMEM supplemented with 20 mM HEPES (no bicarbonate [$HCO_3^-$]) diminishing insulin sensitivity and reproducibility between experiments compared with media buffered with $HCO_3^-$ (see the Materials and Methods section, Fig 1E and F). The half-time (t1/2) of GLUT4 trafficking was 7.5 and 9.5 min in response to 100 and 1 nM insulin, respectively (Fig 1G). For reference, previously reported t1/2 s of GLUT4 translocation are GLUT4 construct-dependent, with HA-GLUT4, GLUT4-tdTomato and HA-GLUT4-GFP t1/2 s reported as 2.5, 5.5 and 9 min, respectively, in 3T3-L1 adipocytes treated with 100 nM insulin (Govers et al, 2004; Karylowski et al, 2004; Burchfield et al, 2013).

We next evaluated the effect of inhibiting critical nodes in the insulin signalling pathway on GLUT4 translocation. 3T3-L1 adipocytes were preincubated with inhibitors targeting phosphatidylinositol 3-kinase (PI3K; GDC0941), 3-phosphoinositide dependent kinase 1 (PDPK1; GSK23344), or Akt (GDC0068 and MK2206) for 10 min, then exposed to 100 nM insulin for 20 min. All inhibitors dose-dependently inhibited insulin-stimulated GLUT4 translocation (Fig 1H and I). Maximal inhibition with GDC0941 was observed at 100 nM, with an IC50 of 8 nM (Fig 1H and I). The other inhibitors all achieved maximal inhibition of insulin-stimulated GLUT4 translocation at 10 µM with IC50's of 289 nM for GSK23344, 121 nM for GDC0068, and 183 nM for MK2206 (Fig 1H and I). A major advantage of using an imaging approach to study endogenous GLUT4 translocation is that it lends itself toward the single cell responses that make up the population response. Consistent with our previous studies of overexpressed GLUT4 (Burchfield et al, 2013; Norris et al, 2021), 100 nM insulin induced a highly heterogeneous response between cells, as evidenced by a broader and flatter population histogram (Fig 1I). Each inhibitor studied dose-dependently reversed this response, with the population response at doses causing maximal inhibition indistinguishable from unstimulated/basal cells (Fig 1I).

Overall, these data support the use of the antibodies targeting exofacial epitopes to study insulin-stimulated endogenous GLUT4 translocation to the PM (Tucker et al, 2018) in both mouse and human adipocytes, although the optimum antibody may be cell line-dependent.

### Studying endogenous GLUT4 translocation in insulin resistant adipocytes

A major reason for developing a system for analysing endogenous GLUT4 trafficking is to study this process in pathophysiological states such as insulin resistance. Interventions that recapitulate causes of insulin resistance in humans, including hyperinsulinemia

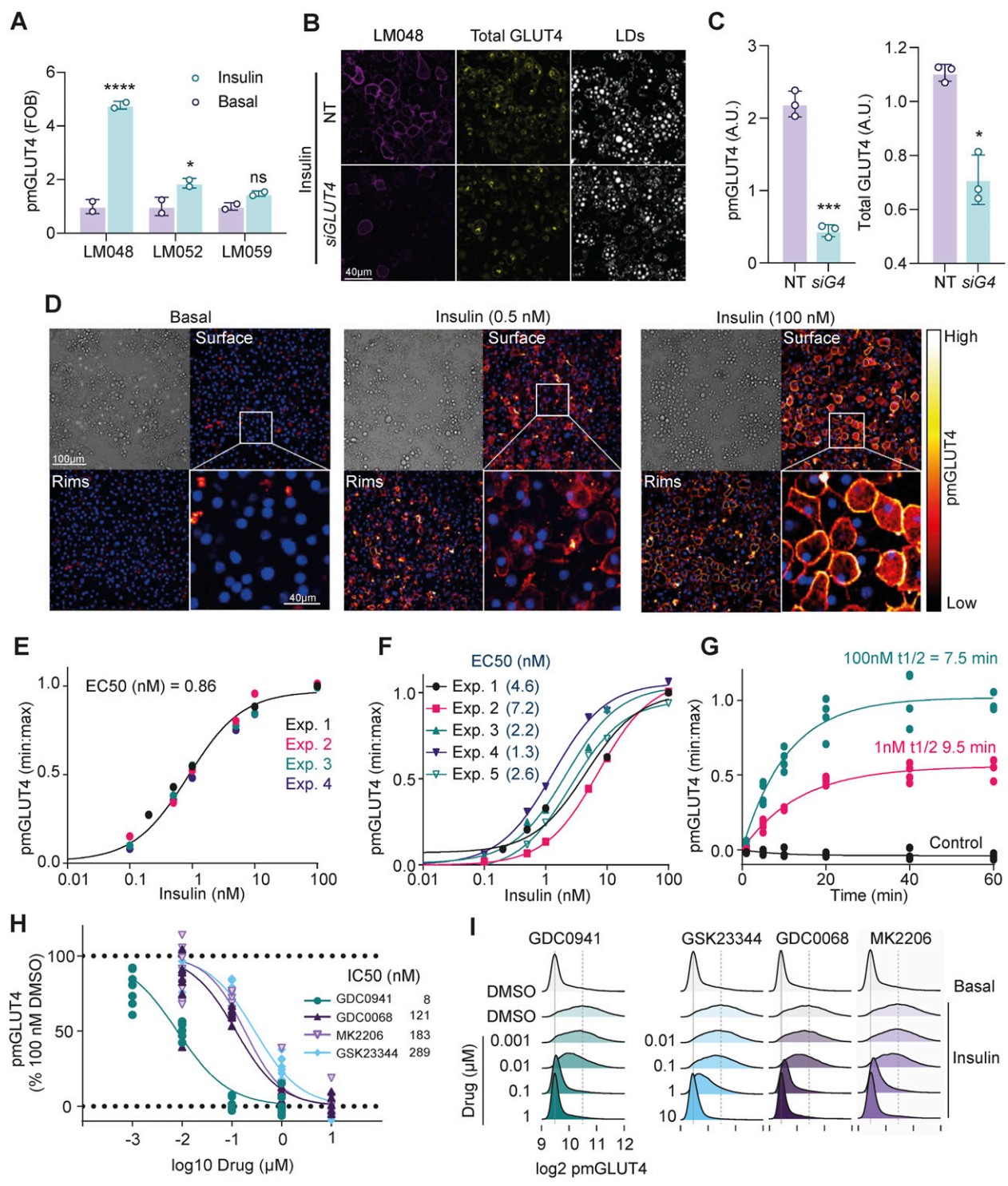

**Figure 1. The anti-GLUT4 exofacial antibody LM048 is ideal for studying the insulin stimulated trafficking of endogenous GLUT4 to the plasma membrane.**
**(A)** Quantification of plasma membrane GLUT4 staining (PM GLUT4) of non-permeabilized mouse 3T3-L1 adipocytes stimulated with or without 100 nM insulin for 20 min and stained with the antibodies LM048, LM052, and LM059 imaged by confocal microscopy (see Fig S1A for representative images). Data are presented as fold-increase in fluorescent signal relative to unstimulated (basal) adipocytes for each antibody. FOB: Fold over basal. Error bars = SD, n = 2. *P < 0.05, ****P < 0.0001 by two-way ANOVA with Dunnett's multiple comparisons test. **(B)** To demonstrate the specificity of the anti-GLUT4 antibody LM048, cells were treated with a non-targeting siRNA (NT) or siRNA targeting *Slc2a4*/*Glut4*, stimulated with 100 nM insulin (20 min), stained with LM048 followed by permeabilization and further staining with an antibody against total GLUT4. Representative confocal images of surface and total staining (scale bars = 40 μm). LDs = lipid droplets. **(C)** Quantification of surface and total GLUT4 labelling in adipocytes treated with non-targeting (NT) siRNA or siRNA targeting GLUT4 (siG4). Error bars = SD, n = 3. *P < 0.05, ***P < 0.001 by unpaired *t* test with Welch's correction. **(D)** Representative bright-field and confocal images of 3T3-L1 adipocytes treated with indicated doses of insulin and stained with the anti-GLUT4 antibody LM048, and DAPI. Images at the bottom surface and midplane (Rims) are presented (scale bars = 100 μm, scale bar of crop = 40 μm). **(E, F)** Quantification of PM GLUT4 content in

(chronic incubation with insulin, CI), chronic inflammation (TNF-$\alpha$) and Cushing's syndrome/inflammatory steroids (dexamethasone, DEX) have been used to model insulin resistance in 3T3-L1 adipocytes. Most of these studies have used GLUT4 reporter constructs to study GLUT4 translocation in insulin resistant cells (Hoehn et al, 2008, 2009; Fazakerley et al, 2018). We next analysed trafficking of endogenous GLUT4 in these insulin resistance models.

3T3-L1 adipocytes were treated with CI (10 nM every 4 h for 24 h), TNF (2 ng/ml for 4 d), or DEX (20 nM for 8 d). Each treatment reduced insulin-stimulated endogenous GLUT4 translocation in response to a maximal insulin dose (100 nM) (Fig 2A). However, only CI and TNF impaired insulin responses at submaximal insulin (1 nM) (Fig 2A). These data are consistent with previous reports where glucose uptake was used as the readout of insulin sensitivity (Tan et al, 2015). Notably, the extent of insulin resistance in response to both CI and TNF was considerably greater than previously described using 3T3-L1 adipocytes or L6 myotubes expressing the HA-GLUT4 reporter (50–60%) (Hoehn et al, 2008, 2009). We speculate this difference stems from two sources. First, it may be that GLUT4 overexpression is somewhat protective. Second, DEX, CI or TNF treatments lower endogenous GLUT4 protein levels and *Slc2a4*/*Glut4* mRNA expression (Fazakerley et al, 2018), which may not translate to overexpressed GLUT4 reporter constructs since these are not under the control of endogenous promoters. Treatment with CI, TNF, or DEX decreased endogenous GLUT4 abundance by 54%, 52%, or 26%, respectively (Fig 2B). GLUT4 down-regulation is a common characteristic of insulin resistance in mouse, rat and human adipose tissue (Garvey et al, 1991; Stephens et al, 1997), so this is likely an important aspect of the insulin resistant state not captured by GLUT4-overexpressing systems.

### Optimizing insulin resistance models for studying endogenous GLUT4 trafficking

As described above, the CI, TNF and DEX models caused a relatively extreme insulin resistance phenotype, likely because these were developed for cells overexpressing GLUT4. To optimise the CI and TNF models for use in wild type cells we exposed 3T3-L1 adipocytes to a range of CI and TNF concentrations for 24 h and assessed both GLUT4 abundance and insulin-stimulated GLUT4 translocation. Both CI (IC50 s = 0.17 nM at 1 nM insulin; 0.71 nM at 100 nM insulin) and TNF (IC50 s = 0.18 ng/ml at 1 nM insulin; 0.55 ng/ml at 100 nM insulin) dose-dependently inhibited insulin-induced GLUT4 translocation (Fig 2C and E). Single cell analysis revealed a heterogeneous insulin response in control cells, indicated by a broad response distribution (Fig 2D and F). Despite this heterogeneous response, as the CI and TNF concentrations increased, the entire population distribution shifted left towards the basal response at both 0.5 and 100 nM insulin (Fig 2D) indicating that insulin

resistance is driven by a concerted response across the population, not by a subset of cells becoming insulin resistant.

Based on these data, we propose exposing 3T3-L1 adipocytes to 0.5 nM insulin and 0.5 ng/ml TNF for 14 h as optimal models of insulin resistance induction. These concentrations of insulin and TNF are between the IC50s of the high and low dose insulin stimulus and resulted in a left shift in the cell population (such that it can be expected that most cells will be affected by the intervention). These are physiologically relevant doses, suggesting that these models will better recapitulate insulin resistance observed in vivo (Parks et al, 2015; da Costa et al, 2016). Representative images of surface GLUT4 in insulin resistant cells are shown in Fig 2I. Twenty-four h treatment with 0.5 nM CI or 0.5 ng/ml TNF also reduced glucose uptake upon insulin stimulation (Fig S2). These lower concentrations of CI and TNF impaired cell surface delivery of GLUT4 without a significant effect on GLUT4 protein levels (Fig 2G and H). These data suggest that the primary defect in insulin resistance is impaired GLUT4 translocation, and that loss of GLUT4 protein is not necessary for the development of insulin resistance. Together, these results demonstrate the utility of assaying endogenous GLUT4 translocation to study insulin resistance in cell culture models.

### siRNA-based workflow to identify regulators of GLUT4 trafficking

Combining measures of endogenous GLUT4 translocation with specific genetic interventions would provide a powerful workflow for identifying regulators of GLUT4 translocation. Using a reverse transfection approach (Kilroy et al, 2009) we observed efficient knockdown of a range of siRNA targets (Fig 3A). Notably, knockdown of *Rab10*, a critical protein in GLUT4 traffic, inhibited endogenous GLUT4 translocation (Fig 3B and C), consistent with earlier findings (Sano et al, 2007). To explore novel regulators of GLUT4 we next assessed the effect of knocking down *Kif13A*, a member of the kinesin family of microtubule motor proteins. Kif13A is of interest because it interacts with Rab10 (Etoh & Fukuda, 2019); it localises to tubular endosomes and is crucial for their formation (Delevoye et al, 2014; Etoh & Fukuda, 2019); and it is phosphorylated on several sites in response to insulin (Humphrey et al, 2013). Depletion of positive controls *Irs1/2* and *Rab10* blunted insulin-induced GLUT4 translocation to the PM (Fig 3C). Similarly, *Kif13A* knockdown reduced GLUT4 at the PM by 60% and 63% after submaximal and maximal doses of insulin, respectively, without affecting cell surface GLUT4 in unstimulated cells (Fig 3C). These data implicate *Kif13A* as a regulator of insulin-responsive GLUT4 translocation and highlight the efficacy of combining siRNA knockdown in differentiated adipocytes with assessment of endogenous GLUT4 translocation for discovering new regulators of GLUT4 trafficking.

---

response to specified insulin doses in either bicarbonate (E) or HEPES (F) buffered media. Data points from individual experiments were fitted to a 3-parameter Hill equation. The EC50 for cells in bicarbonate buffer is indicated on the graph. For HEPES buffered media, a single curve did not adequate fit all datasets using Akaike's Information Criterion. **(G)** Quantification of PM GLUT4 using the anti-GLUT4 antibody LM048 in adipocytes treated with the indicated doses of insulin for specified times. Data are presented as fluorescence intensity of pmGLUT4 signals a percentage of the maximum response n = 4. **(H, I)** Population (H) and single cell (I) plasma membrane endogenous GLUT4 abundance data from wild-type 3T3-L1 adipocytes treated with or without 100 nM insulin in combination with DMSO (control), the PI3K inhibitor GDC0941, Akt inhibitors GDC0068 and MK2206, or PDK1 inhibitor GSK23344. Cells were stained with the LM048 antibody against endogenous GLUT4 in non-permeabilized cells and expressed as a proportion of DMSO-treated controls n = 6.

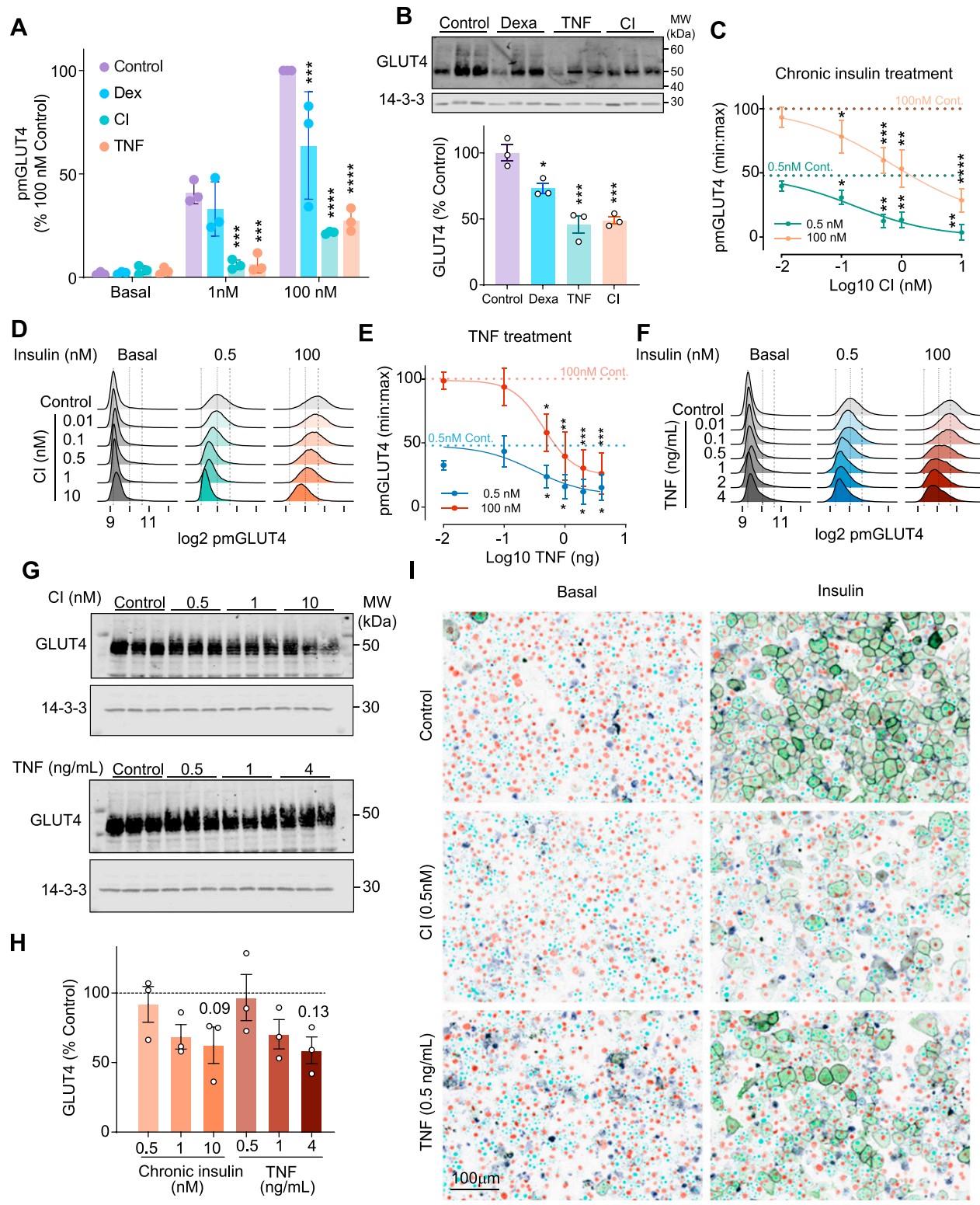

**Figure 2. Studying insulin-stimulated translocation of endogenous GLUT4 in insulin resistant adipocytes.**
**(A)** PM GLUT4 in 3T3-L1 adipocytes stimulated with 1 or 100 nM insulin after treatment with either medium, dexamethasone (DEX; 20 nM for 8 d), TNF-α (TNF; 2 ng/ml for 4 d) or chronic insulin (CI; 10 nM every 4 h for 24 h). Data are presented as mean ± SD. ****$P < 0.0001$, *$P < 0.05$ versus basal by two-way ANOVA with Šidák's multiple comparisons test. **(B)** Total cellular GLUT4 was assessed by Western blot (upper panel). Each track represents an independent biological replicate, from left to right control, Dexamethasone, TNF and CI. Quantitation (lower panel) of GLUT4 normalized to control cells (set to 100%). Data are mean ± SD, n = 3 with *$P < 0.05$ compared with control by 1-way ANOVA with Dunnet's test for multiple comparisons. **(C)** Dose response with CI over 24 h. Data are min:max normalized to the control basal and 100 nM response

## A high-content screening platform to identify regulators of insulin-regulated membrane traffic and lipid content in adipocytes

The combination of siRNA-mediated gene depletion and assessment of endogenous PM GLUT4 provided the opportunity to establish a high-content screening platform for genetic and small molecule screening. However, we reasoned that such an approach would benefit from additional relevant phenotypic measures and a more efficient imaging and analysis workflow.

First, to expand the utility of our assay we included the simultaneous analysis of other trafficking proteins such as the TfR, which moves to the cell surface in response to insulin via a distinct trafficking pathway to GLUT4 (Habtemichael et al, 2011). Using an antibody that recognizes an extracellular epitope on TfR, we simultaneously assessed PM abundance of endogenous TfR and GLUT4 in the same cells (Lyons et al, 2009). siRNA-mediated knockdown of *TfR* depleted the TfR surface signal (Fig 4A and B) and total cellular TfR (Fig 4C), demonstrating the specificity of this approach. Consistent with previous studies, we observed that cell surface levels of TfR under basal conditions were much higher (35% of maximal stimulated levels) than GLUT4 (6.5% of maximal stimulated levels) (Habtemichael et al, 2011). Moreover, TfR underwent insulin-dependent movement to the PM and this occurred with a shorter half time (3 min) than observed for endogenous GLUT4 (9 min; Fig 4D). The insulin dependent movement of TfR was sensitive to PI3K/Akt inhibitors (Fig S3A, right panel).

Second, we added measures of cell number using nuclei counting to control for interventions that affect cell viability (Fig 6D and E). Finally, we imaged lipid droplets using digital phase contrast microscopy (Fig 5) This provided measures of lipid content using a non-destructive, label free method. This digital phase contrast-based method detected increased cellular lipid content during adipocyte differentiation (Fig 5A–C), and lower lipid droplet volume and fewer lipid droplets in adipocytes depleted of the lipid droplet regulators *Plin1* or *Cidec* (de la Rosa Rodriguez & Kersten, 2017) (Fig 5D–F). Together, the inclusion of both measures of PM TfR and lipid droplet features provides a powerful platform for studying protein trafficking, adipocyte differentiation and lipid storage.

### High-content screen for regulators of insulin-stimulated membrane traffic and lipid droplets

To test combining siRNA knockdown with our high-content imaging workflow we selected a series of known regulators of insulin-stimulated GLUT4 or TfR traffic as "genes-of-interest." These were *Akt1/2* (Cong et al, 1997), *Tbc1d4* (Sano et al, 2003), *Rab10* (Sano et al, 2007; Chen et al, 2012), *Lnpep (IRAP)* (Larance et al, 2005; Jordens et al, 2010), *Stx6* (Perera et al, 2003; Watson et al, 2008), *Stx16* (Proctor et al, 2006), *Vps45* (Burkhardt et al, 2008; Roccisana et al, 2013), *Cltc* (Vassilopoulos et al, 2009; Fumagalli et al, 2019), *Rab14* (Reed et al, 2013), or *Trarg1* (Fazakerley et al, 2015). All siRNAs appeared to be well tolerated as we observed no major loss of cells (Fig 6C). All known GLUT4 regulators altered PM GLUT4 with the exception of *Akt1/2*, *Lnpep*, and *Cltc* (Fig 6A). Knockdown of *Tbc1d4*, *Rab10*, *Trarg1*, *Stx6*, *Stx16*, and *Vps45* impaired GLUT4 translocation responses to both submaximal and maximal doses of insulin (Fig 6A). Of these, *Rab10*, *Stx16*, and *Trarg1* had the largest effects. These data are consistent with previous reports with the exception of *Tbc1d4/As160* where knockdown would be expected to increase PM GLUT4. This discrepancy may be explained by lower GLUT4 abundance because *Tbc1d4/As160* knockout lowers cellular GLUT4 by ~50% (Chadt et al, 2015). *Rab14* knockdown potently increased cell surface GLUT4 content in response to insulin (Fig 6A). *Rab14* has been implicated in the intracellular sorting of GLUT4 from endosomes to the PM or GSVs, and previous knockdown studies have reported decreased PM GLUT4 in response to lower *Rab14* expression (Chen et al, 2012; Reed et al, 2013; Brewer et al, 2016). However, the same inhibitory phenotype was observed in both *Rab14* knockdown and overexpression (Reed et al, 2013), suggesting that differences in the extent (and length of time) of *Rab14* knockdown may have distinct effects on GLUT4 traffic.

*Rab10*, *Trarg1*, *Stx6*, and *Stx16* knockdown also impaired TfR responses (Fig 6B). This suggests that these proteins may participate in a common trafficking step for both GLUT4 and TfR. Depletion of *Trarg1* most strongly impaired TfR cell surface abundance. Interestingly, a number of these knockdowns affected lipid droplets, with *Rab14* depletion having the greatest effect to lower lipid content and droplet number (Fig 6D and E). *Trarg1* depletion also increased lipid droplet number (Fig 6E) despite not affecting the overall lipid content within cells (Fig 6D), placing Trarg1 as a regulator of TfR traffic and lipid droplets in addition to GLUT4 traffic.

We used this high-content assay to find interactors of Trarg1 that may contribute to the regulation of GLUT4, TfR or lipid droplet morphology by Trarg1. An unbiased immunoprecipitation-mass spectrometry study revealed Bcl9l as a Trarg1 interactor. Bcl9l was identified with high sequence coverage (61 peptides; ~50% sequence coverage), was the most enriched interactor (Table S1, volcano plot of positive interactors is shown in Fig S3C), and the Trarg1-Bcl9l interaction decreased in response to insulin (Fig S3B, C, and E and Table S1). Although Bcl9l has not been implicated in GLUT4 or trafficking before, it is a transcriptional co-activator of

---

and are presented as the population mean ± SD of n = 6 experiments. *P < 0.05 by mixed effect model with Geisser-Greenhouse Correction and Dunnet's multiple comparisons test. **(D, E, F)** Single cell analysis of surface GLUT4 staining using LM048 in 3T3-L1 cells after CI or TNF treatment for 24 h, ±1 or 100 nM insulin. **(C, D)** Histogram displaying single cell responses from an average of 25,611 cells per indicated condition (C). Dotted lines represent the peak of the control basal, 0.5 and 100 nM responses. (E) Dose response with TNF over 24 h. Data are min:max normalized to the control basal and 100 nM response and are presented as the population mean ± SD of n = 6 experiments. *P < 0.05, **P < 0.01, ***P < 0.001, ****P < 0.0001 by mixed effect model with Geisser–Greenhouse Correction and Dunnet's multiple comparisons test. **(D)** and **(F)** were run simultaneously and they share the control cells, the control cells have been plotted twice for clarity. **(E, F)** Histogram displaying single cell responses from an average of 21,456 cells per indicated condition (E). Dotted lines represent the peak of the control basal, 0.5 and 100 nM responses. **(G)** Total GLUT4 was assessed after CI (left) or TNFα (right treatment 14-3-3 was used as loading control [bottom panel]). **(H, I)** Densitometric analysis of GLUT4 abundance from (I), ± SD of n = 3 experiments. P-values as indicated by one-way ANOVA with Dunnet's test for multiple comparisons. **(I)** Representative image of 3T3-L1 adipocytes in basal (left panel) and after 0.5 nM insulin stimulation (right panel) in control and insulin resistance cells. Green = PM-GLUT4, orange = nucleus, cyan = lipid droplets.

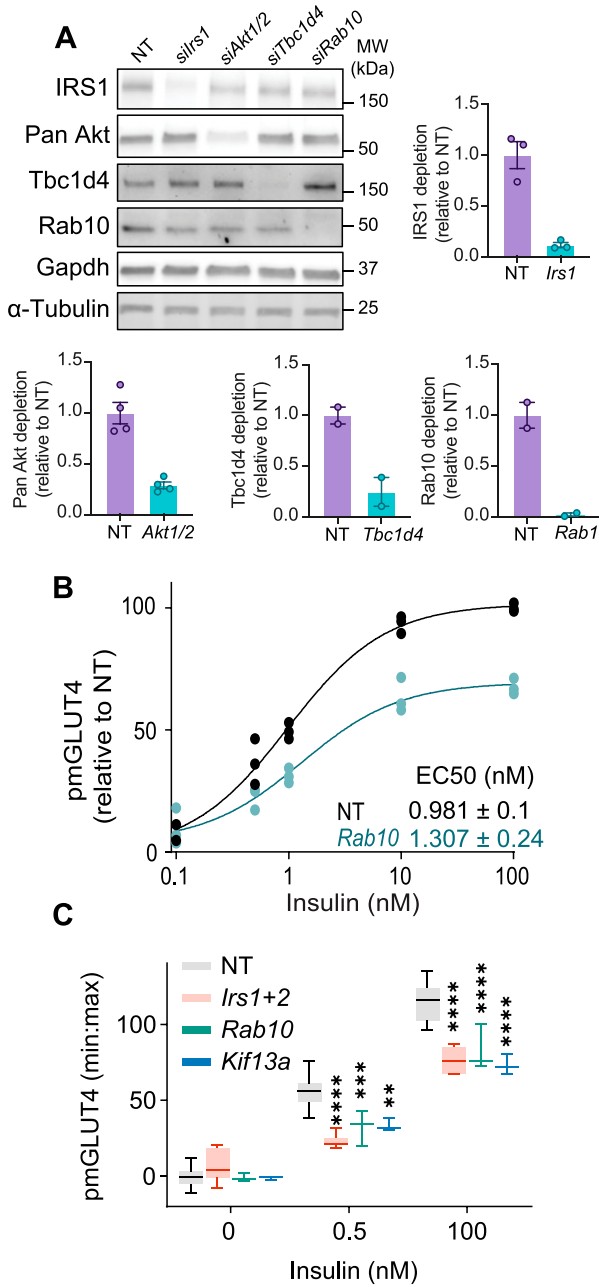

**Figure 3. siRNA-based workflow to identify regulators of GLUT4 trafficking. (A)** siRNA-mediated knockdown of known GLUT4 trafficking mediators in fully differentiated 3T3-L1 adipocytes and the densitometry of the blots n = 2–4. **(B)** Quantification of PM GLUT4 abundance (using LM048) in cells treated with control siRNA or siRNA targeting Rab10 in response to varying doses of insulin n = 3. EC50s of each curve are expressed on the graph. **(C)** Quantification of PM GLUT4 abundance (using LM048) in cells treated with control siRNA or siRNA targeting IRS1/2, Rab10 or Kif13A in response to 0.5 and 100 nM insulin (data presented as mean ± SD of n = 4 experiments). ***$P < 0.001$, ****$P < 0.0001$ by mixed effect model with Geisser–Greenhouse Correction and Dunnet's multiple comparisons test.

β-catenin, which has been implicated in insulin sensitivity in skeletal muscle (Masson et al, 2020). Knockdown of *Bcl9l* (Fig S3D and E) did not affect GLUT4 (Fig 6A), but increased PM TfR in cells treated with 100 nM insulin (Fig 6B) and also increased lipid droplet

number per cell (Fig 6E). Overall, our combined analysis of GLUT4 and TfR traffic have revealed that Trarg1 is a more general regulator of insulin-regulated protein trafficking than GLUT4 alone, as well as a regulator of lipid droplets. Furthermore, the Trarg1-Bcl9l interaction may play a role in Trarg1-dependent TfR traffic and in linking Trarg1 to lipid droplet number. These new observations on the role of Trarg1 in adipocyte biology highlight the utility of assessing TfR responses and lipid droplet features alongside GLUT4.

### Endogenous GLUT4 appears more sensitive to gene knockdown compared with GLUT4 overexpressing cells

We next compared the responses of endogenous GLUT4 and overexpressed HA-GLUT4-mRuby3 (HA-GLUT4-mR3) with gene depletion. Western blotting revealed an overexpression of HA-GLUT4-mR3 that was similar to the level endogenous GLUT4, indicating that these cells contain roughly twice the amount of total GLUT4 as wild-type cells (Fig S5A). Subcellular fractionation revealed a highly similar distribution of HA-GLUT4-mR3 to endogenous GLUT4 under basal and insulin-stimulated conditions (Fig S5A). This was supported by surface labelling of GLUT4 in response to 100 nM insulin (using the LM048 antibody), whereby the PM GLUT4 signal was ~2 times greater in HA-GLUT4-mR3 cells compared with WT cells (Fig S5B). We performed the same series of knockdowns as in Fig 6 in adipocytes expressing HA-GLUT4-mR3, and correlated surface GLUT4 and TfR levels in each cell line under basal or insulin-stimulated conditions across all knockdowns (0.5 and 100 nM insulin).

Under basal conditions, there was little effect of knockdown on surface GLUT4 levels in either cell line and unsurprisingly, there was no correlation between them ($r^2 = 0.01$, slope = -0.17) (Fig 7A). In response to either submaximal or maximal insulin concentrations, the impact of knockdown on surface GLUT4 levels in HA-GLUT4-mR3 cells was weaker than that of endogenous GLUT4, as demonstrated by the relatively flat slopes of the correlation (slope = 0.16 for both submaximal and maximal insulin) and the low correlation coefficient ($r^2 = 0.08$ for both submaximal and maximal insulin) (Fig 7C and E). In contrast, the correlation between cell lines was substantially higher for surface TfR under basal conditions ($r^2 = 0.3$, $P < 0.0001$, Fig 7B), and upon insulin stimulation ($r^2 = 0.45$, $P < 0.0001$ for submaximal and $r^2 = 0.47$, $P < 0.0001$ for maximal insulin) with a broadly equivalent effect size between wild-type and HA-GLUT4-mR3 -overexpressing cells (Slope = 1.02 and 0.88 for submaximal and maximal insulin concentration, respectively) (Fig 7D and F). The distinct GLUT4 and HA-GLUT4-mR3 responses to gene knockdown are unlikely explained by differences in knockdown efficiency between cell lines, given that the effects on TfR are so consistent and that no difference was observed in the depletion of Akt1/2 and Rab10 between the cell lines (Fig S4). Therefore, these data suggest that GLUT4 overexpression has a specific effect on the sensitivity of GLUT4 trafficking responses to gene knockdown. One notable case where HA-GLUT4-mR3 cells exhibited a greater effect for both HA-GLUT4-mR3 and TfR was for *Akt1/2* knockdown at 1 nM (Fig 7C and D). This is interesting given that Akt is a central node in the insulin signalling pathway and these data may indicate change in insulin sensitivity or network rewiring as a result of GLUT4 overexpression. To further explore this, we tested the sensitivity of both WT and HA-GLUT4-mR3 overexpressing lines to the Akt inhibitor MK2206 at

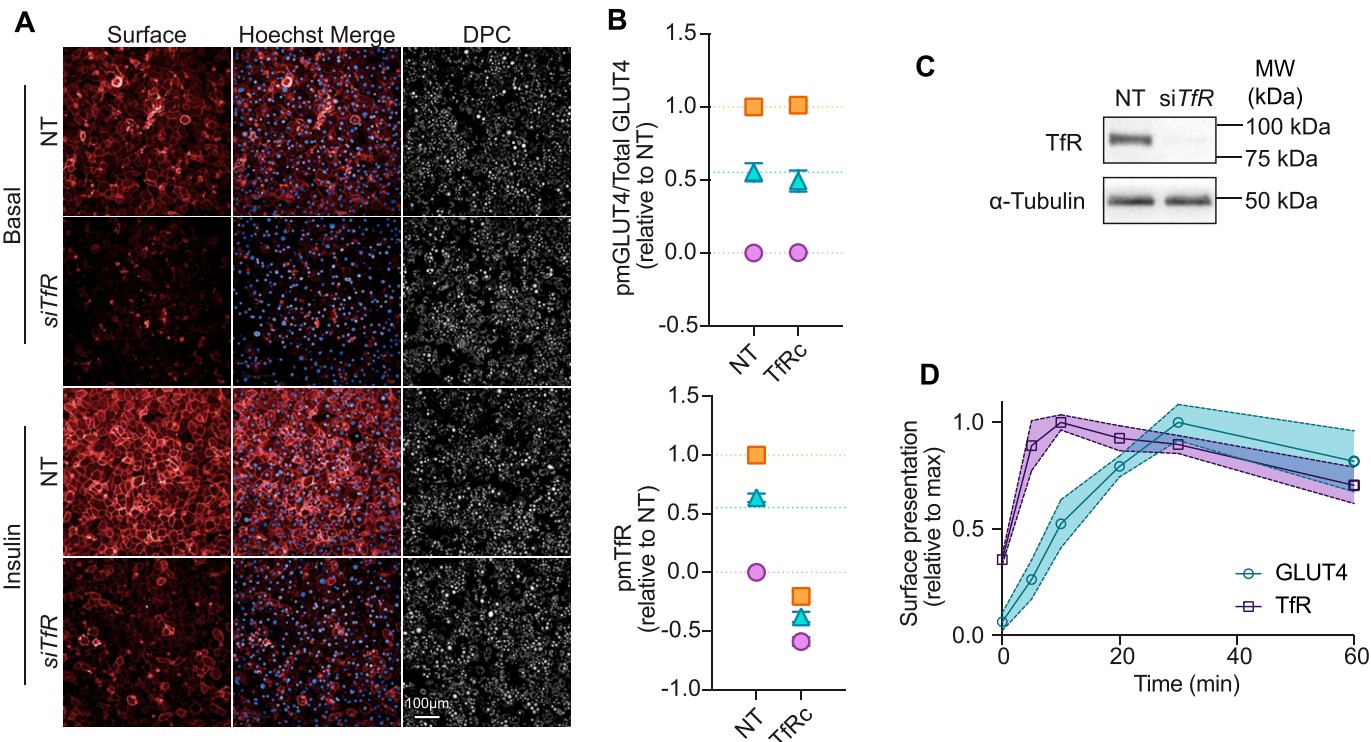

**Figure 4. Assessment of insulin-stimulated TfR translocation in 3T3-L1 adipocytes.**
**(A)** Surface labelling of TfR in non-permeabilized cells, treated with either non-targeting siRNA (Ctrl) or siRNA targeting Transferrin Receptor (siTfR), in the presence and absence of 100 nM insulin. **(A, B)** Quantification of GLUT4 (upper) and TfR trafficking responses (lower) of cells in (A). **(C)** Western blot of total cellular TfR in lysates from cells treated with non-targeting siRNA (NT) or siRNA targeting TfR (siTfR). Membranes were blotted with antibodies recognizing TfR or α-Tubulin as a loading control. **(D)** Time-course of surface Transferrin Receptor (TfR) and GLUT4, using LM048, in 3T3-L1 adipocytes in response to 100 nM insulin (data presented as mean ± SD of n = 3 experiments).

submaximal (1 nM) and maximal (100 nM) insulin. No difference in sensitivity to MK2206 was observed in insulin-induced GLUT4 translocation between WT and HA-GLUT4-mR3 cells (Fig S5C and D), suggesting that the differences in response to *Akt1/2* knockdown between WT and GLUT4 overexpressing may be due to altered responses to prolonged lower expression of Akt isoforms, rather than the sensitivity of GLUT4 to Akt activity, or differences in KD efficiency when these experiments were performed. To-gether, these analyses highlight key differences in the way that WT and GLUT4-overexpressing cells respond to gene knockdown and acute kinase inhibition, and the enhanced sensitivity afforded by studying endogenous GLUT4 trafficking over tradi-tional assays that rely on overexpression of GLUT4 reporter constructs.

## Discussion

GLUT4 translocation to the PM in muscle and fat is required for whole body glucose homeostasis (James et al, 2021). Despite its importance, tools for studying endogenous GLUT4 trafficking have been lacking. Here we use new antibodies against the exofacial domain of GLUT4 (Tucker et al, 2018) to develop an imaging-based assay for assessing endogenous GLUT4 trafficking in cultured murine and human adipocytes (Fig 1). We developed new models of insulin resistance, which revealed a disconnect between impaired

GLUT4 trafficking and GLUT4 protein down-regulation in insulin resistance (Fig 2). In addition, our high-content assay can be (1) multiplexed with additional measures such as cell surface TfR abundance and lipid droplet features and (2) combined with pharmacological or genetic approaches (Figs 1 and 6) to study the effects of small molecules or gene knockdown on insulin-responsive GLUT4 and TfR translocation and adipocyte lipid storage. This approach revealed that Kif13A is a novel regulator of GLUT4 traffic (Fig 3), that Trarg1 regulates GLUT4 and TfR traffic as well as lipid droplet size, and that the Trarg1-interactor Bcl9l is a novel regulator of TfR traffic and lipid droplets in adipocytes (Fig 6). Compared with similar workflows that rely on GLUT4-overexpressing systems, this assay demonstrated greater sensi-tivity for detecting genetic and pharmacological perturbants of GLUT4 traffic.

Expression of epitope-tagged GLUT4 reporter constructs has allowed measures of GLUT4 PM content. Until recently, this has not been possible for endogenous GLUT4. Here we identify optimal antibodies for labelling of endogenous GLUT4 (Tucker et al, 2018) within the PM of murine 3T3-L1 adipocytes (a leading model for studying insulin sensitivity and GLUT4 traffic) and SGBS human adipocytes (Fig 1). Of note, we found that none of the exofacial antibodies tested significantly labelled endogenous GLUT4 in C2C12 and L6 myotubes (data not shown). This likely reflects low ex-pression of GLUT4 in these lines compared with adult tissue. The LM048 antibody was reported to be state-dependent for human

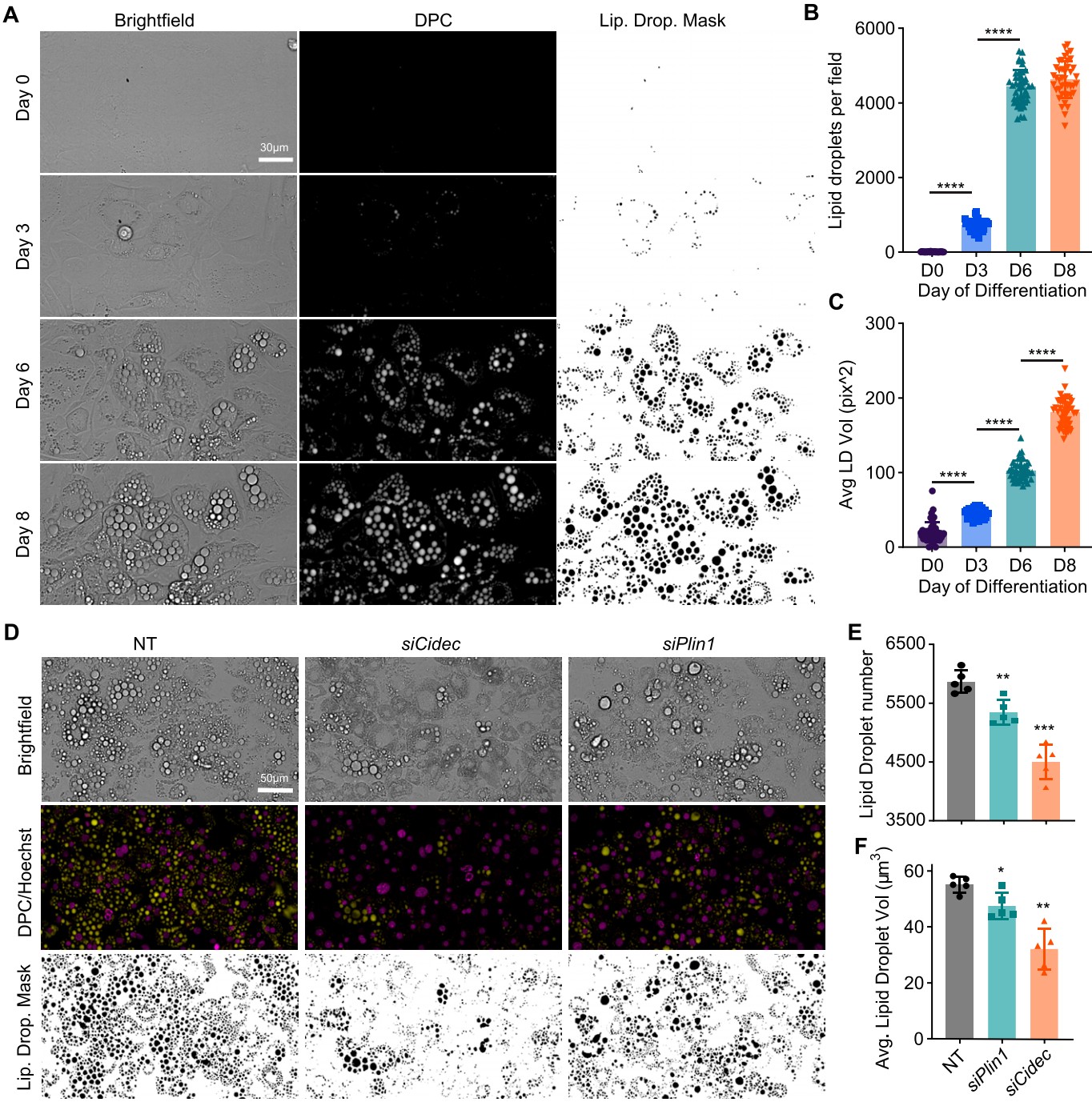

**Figure 5. Validation of digital phase contrast (DPC) imaging to measure lipid droplet content in 3T3-L1 adipocytes.**
**(A)** Bright-field and DPC images of 3T3-L1 cells throughout differentiation from preadipocytes (day 0) to adipocytes (day 8), and the resultant lipid droplet mask used to quantify lipid accumulation. **(A, B, C)** Number of lipid droplets (B) and average lipid droplet volume (C) observed throughout the differentiation time-course in (A) (Data are presented as mean + SD, n = 2, 40 wells per experiment ****$P$ < 0.001, by one-way ANOVA with Dunnet's test for multiple comparisons). **(D)** The effect of siRNA-mediated knockdown of *Cidec* and *Plin1* on lipid droplets in differentiated 3T3-L1 adipocytes, as measured by brightfield and DPC in comparison with a non-targeting siRNA control. Hoechst 33342 was used as a control for cell number/density. **(D, E, F)** Number of lipid droplets (E) and average lipid droplet volume (F) observed for the cells in (D) (Data is presented as mean + SD, n = 5, ****$P$ < 0.001, by one-way ANOVA with Dunnet's test for multiple comparisons).

GLUT4, favouring the outward confirmation (Tucker et al, 2018). Although we have not tested this directly, we performed all fixation or antibody staining in assays using this antibody in the absence of glucose, which would likely promote this outward confirmation and

favour antibody binding. However, one limitation of using this LM048 antibody is that experimental interventions that alter GLUT4 conformation might affect antibody binding and thereby falsely report changes in PM GLUT4. Nevertheless, these antibodies

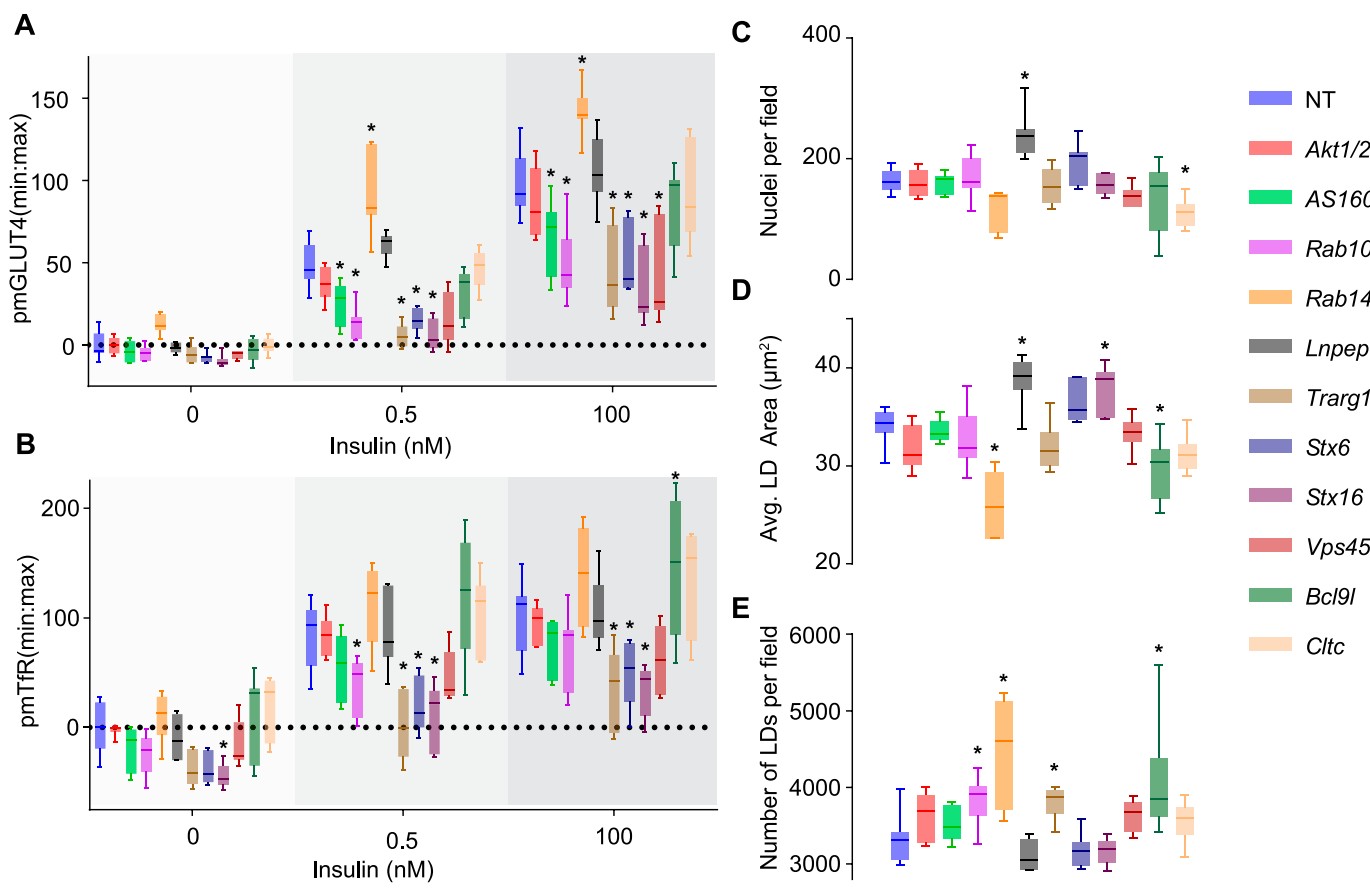

**Figure 6. High-content screen for regulators of insulin-stimulated membrane traffic and lipid droplet content.**
**(A, B)** Quantification of fluorescently labelled surface GLUT4 (using LM048) (A) and TfR (B) in response to 0.5 and 100 nM insulin after knockdown of a panel of regulators of GLUT4 trafficking, in non-permeabilized 3T3-L1 adipocytes. **(A, B, C)** Number of nuclei per field of view in the cells in (A, B). **(A, B, D)** Average lipid droplet area for cells in (A, B). **(A, B, E)** Number of lipid droplets for the cells in (A, B). Legend on the right indicates the siRNA conditions in each graph. Data presented as mean ± SD, n = 4–8, *$P <$ 0.05, by mixed effect model with Geisser–Greenhouse Correction and Dunnet's multiple comparisons test.

represent an advance for studying endogenous GLUT4 trafficking and may offer means to study PM appearance of endogenous GLUT4 in vivo or ex vivo, including muscle tissues.

Insulin resistance is characterised by attenuated insulin induced-GLUT4 trafficking (Hoehn et al, 2008, 2009; Fazakerley et al, 2018) and GLUT4 down-regulation in adipose tissue (Garvey et al, 1991; Stephens et al, 1997). The latter may not be reproduced in GLUT4-overexpressing systems, thereby obscuring the relationship between GLUT4 down-regulation and impaired GLUT4 translocation in adipocyte insulin resistance. This new approach for studying endogenous GLUT4 allowed us to optimize models of insulin resistance that require more physiological interventions, minimize cell handling and reduce the time required to generate insulin resistance. Using these new models, we observed impaired insulin-stimulated GLUT4 translocation without substantial changes in GLUT4 protein abundance. This suggests that impaired GLUT4 trafficking is either independent or upstream of GLUT4 protein down-regulation in insulin resistance. For example, it may be that aberrant GLUT4 traffic and localisation, as observed in insulin resistance (Garvey et al, 1993; Vassilopoulos et al, 2009), leads to increased GLUT4 turnover or decreased *Glut4/Slc2a4* expression. Another possibility is that GLUT4 traffic is more sensitive to insults

that induce insulin resistance, and that loss of GLUT4 is only observed in response to higher doses or longer exposures. Using optimized insulin resistance models to study endogenous GLUT4 will improve mechanistic studies into insulin resistance drivers by avoiding potential off-target effects associated with GLUT4 over-expression and perturbations used to induce insulin resistance.

The combination of gene knockdown and imaging-based high-content (approaching high-throughput) monitoring of endogenous GLUT4 translocation allows screening for gene-of-interest effects on GLUT4 in adipocytes. As with all genetic approaches, there may be limitations issues with redundancy that limit the phenotypes observed for single-gene knockdowns. For example, there was a marked disparity between the effect of pharmacological inhibition of Akt (Figs 1H and I and S5) and knockdown of *Akt1* and *Akt2* (Fig 7). This is likely due to a combined effect of isoform redundancy (i.e., Akt3 is still expressed) and the need for relatively little active Akt for maximal GLUT4 translocation (Hoehn et al, 2008) Thus, false negatives might arise from incomplete knockdown, or functional redundancies/compensatory mechanisms for the target gene. Nevertheless, this platform is sensitive to most of the known regulators of GLUT4 traffic (Figs 3 and 6), and revealed the kinesin protein Kif13A as a new regulator of GLUT4 (Fig 3). Recent reports

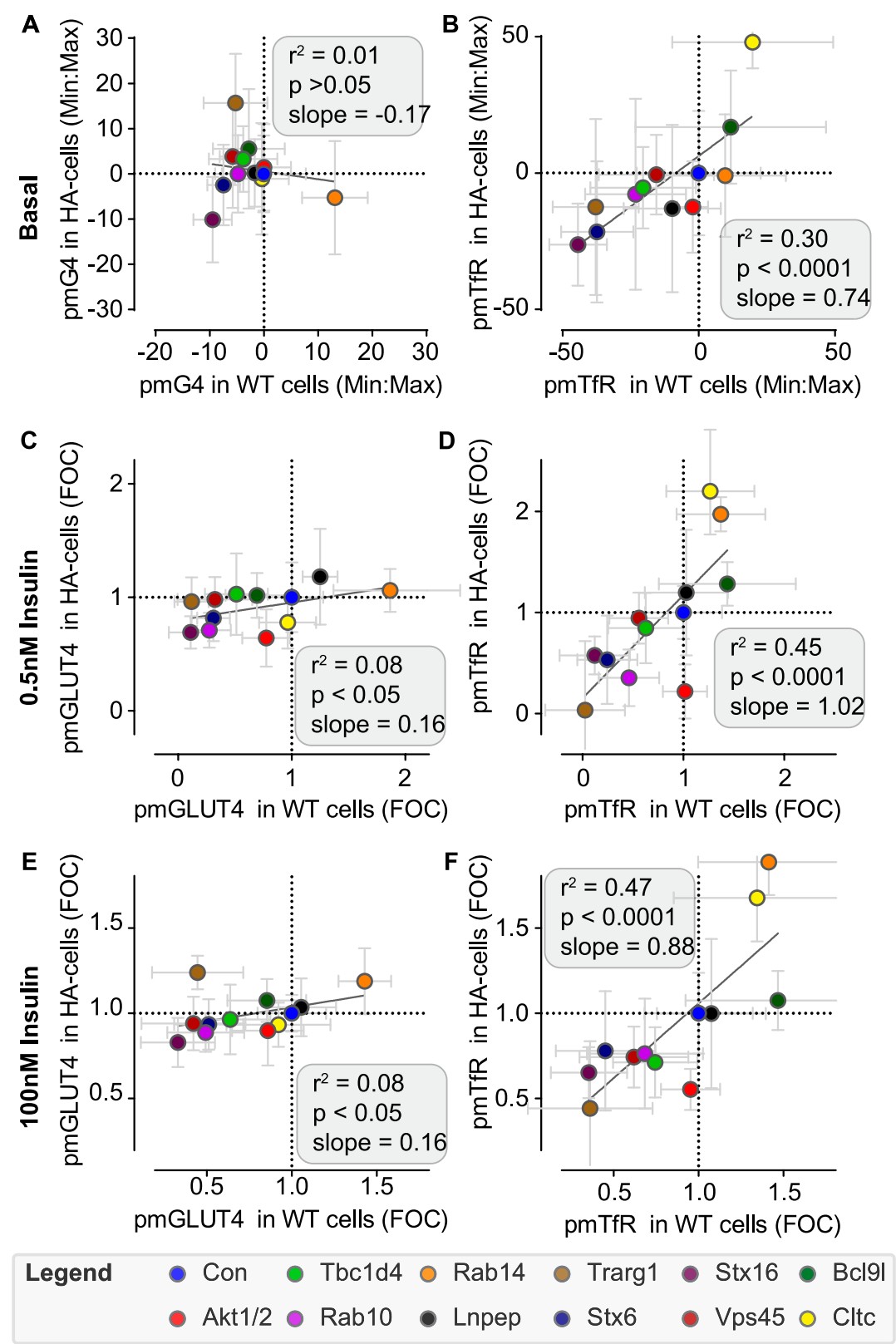

**Figure 7. Trafficking of overexpressed HA-GLUT4-mR3 protects cells from genetic perturbations in comparison with endogenous GLUT4.**
**(A, C, E)** Correlations between the endogenous GLUT4 or HA-GLUT4-mR3 trafficking responses in wild-type and HA-GLUT4-mR3 expressing cells, respectively, in response to siRNA knockdown of several regulators of GLUT4 trafficking. **(B, D, F)** Correlations between the TfR trafficking responses in wild-type and HA-GLUT4-mRuby3-expressing

demonstrated that Kif13A is an insulin-regulated phosphoprotein (Humphrey et al, 2013) that interacts with Rab10 to regulate tubular endosome formation (Delevoye et al, 2014; Etoh & Fukuda, 2019). Therefore, Kif13A may mediate GLUT4-vesicle trafficking on microtubules to the cell periphery as reported for mannose-6-phosphate receptor (Nakagawa et al, 2000). A previous study reported *Kif13A* down-regulation did not affect HA-GLUT4-GFP traffic (Brumfield et al, 2021). We speculate this difference may be due to differences between endogenous and overexpressed GLUT4 traffic. Overall, combining genetic (or pharmacological) interventions with endogenous PM GLUT4 measures represents an important new method for studying GLUT4.

Our studies suggest that GLUT4 overexpression confers some resistance against both genetic perturbations (Fig 7) and treatments that induce insulin resistance. For example, standard treatments to induce insulin resistance resulted in far greater inhibition of endogenous GLUT4 (Fig 2) than observed for GLUT4 reporters (Hoehn et al, 2008; Fazakerley et al, 2018). Furthermore, adipocytes overexpressing GLUT4 were less sensitive to knockdown of known GLUT4 regulators despite near identical results for TfR in the same cells (Fig 7) and no significant differences in knockdown of tested targets between the cell lines (Fig S4). We suggest that there are two possible explanations for these observations. First, overexpression systems may overwhelm saturable steps in traffic, leading to aberrant trafficking (e.g., increased PM GLUT4 in unstimulated cells). For instance, different GLUT4 reporter constructs have apparent differences in their t1/2 s, varying from 2.5 to 9 min (Govers et al, 2004; Karylowski et al, 2004; Burchfield et al, 2013). TfR had faster translocation kinetics than GLUT4, so a faster t1/2 for GLUT4 reporters may indicate saturation of the GSV compartment and mislocalization of GLUT4 to a TfR-positive compartment. Second, increased glucose uptake from long-term GLUT4 overexpression may result in transcriptional changes that render adipocytes less susceptible to insulin resistance. This may be through glucose-responsive transcription factors such as CHREBP, which has been implicated in regulating insulin sensitivity (Herman et al, 2012). Such transcriptional differences may result in signal or trafficking network rewiring that may explain some of the differences observed between these lines.

An advantage of the high-content imaging approach to study PM GLUT4 is that it allowed us to also measure lipid droplet features (Fig 5) and PM TfR (Figs 4 and 6) in parallel. Our data suggest that insulin-stimulated TfR translocation is both PI3K- and Akt dependent, as is GLUT4. Because of the different kinetics of TfR and GLUT4 translocation to the PM, insulin may activate two independent exocytosis compartments, in line with observations that ablation of TfR-positive compartments did not block insulin-induced GLUT4 translocation (Martin et al, 1998). Despite this, only *Vps45* and *Rab14* knockdown selectively affected GLUT4 (Fig 6) suggesting that these proteins act in more GLUT4-specific trafficking pathways. For example, both proteins are implicated in traffic from endosomes to Golgi (Junutula et al, 2004; Rahajeng et al,

2010) On the other hand, *Trarg1*, *Rab10*, *Tbc1d4*, *Stx6*, or *Stx16* knockdowns affected both GLUT4 and TfR traffic. The Rab10/TBC1D4 couple plays a key role in GLUT4 translocation (Sano et al, 2007; Sakamoto & Holman, 2008). Although GLUT4 was more sensitive than TfR trafficking to *Rab10* depletion, suggesting that Rab10 plays a more central role in GLUT4 traffic (Babbey et al, 2006; Chen & Lippincott-Schwartz, 2013), our data suggest that Rab10/Tbc1d4 regulates the fusion of multiple exocytic vesicles with the PM. Together, this suggests overlap in the regulatory machinery that underlies GLUT4 and TfR translocation. Overall, this method offers the opportunity to learn more about insulin-regulated TfR trafficking, and to aid mechanistic studies into specific new regulators of GLUT4.

Trarg1 is required for maximal insulin-stimulated GLUT4 translocation (Beaton et al, 2015; Fazakerley et al, 2015). However, its mechanism-of-action remains unclear. We have identified that *Trarg1* depletion impaired TfR trafficking and increased lipid droplet number. These data suggest that Trarg1 may play a more general role in regulating membrane traffic than GLUT4 alone. Other members of the dispanin family, to which Trarg1 belongs, are implicated in membrane trafficking through a variety of mechanisms including vesicle fusion at the PM (Coleman et al, 2018), membrane fluidity (Rahman et al, 2020), and endolysosomal traffic (Spence et al, 2019). This provides a number of avenues for future mechanistic studies. We also found that the insulin-responsive Trarg1 interactor Bcl9l altered TfR PM abundance and increased lipid droplet number. Although Bcl9l is reported to be predominantly nuclear localised, consistent with its role as a transcriptional co-activator, it is also present in the cytoplasm of ameloblasts and supports the secretion of enamel components (Cantù et al, 2017). Further studies are required to elucidate how Trarg1 and its binding proteins cooperate to regulate TfR trafficking and lipid droplet morphology.

In summary, we have described and validated a new approach to study endogenous GLUT4 traffic and have highlighted the utility of this antibody for high-content screening and analysis of GLUT4 translocation at population and single-cell levels. We envisage that these methods will accelerate work into the fundamental biology of regulated GLUT4 exocytosis and the pathophysiological mechanisms underpinning insulin resistance and type 2 diabetes.

# Materials and Methods

### 3T3-L1 fibroblast culture and differentiation into adipocytes

Mycoplasma-free 3T3-L1 fibroblasts obtained from 3T3-L1 Howard Green (Harvard Medical School) were maintained in DMEM (Gibco by Life Technologies) supplemented with 10% FBS (vol/vol) (Gibco by Life Technologies) and 2 mM GlutaMAX (Gibco by Life Technologies) at 37°C and 10% $CO_2$. 3T3-L1 fibroblasts were differentiated to adipocytes as previously described (Norris et al, 2018; Krycer et al,

---

cells, in response to siRNA knockdown of regulators of GLUT4 trafficking (as indicated). **(A, B, C, D, E, F)** Plasma membrane GLUT4 (pmG4) and TfR were determined under basal (A, B), and 0.5 nM (C, D) or 100 nM insulin-stimulated conditions (E, F). FOC = Fold over control. Correlations are indicated by the respective $r^2$ and *P*-values as determined by simple linear regression.

2020a, 2020b). Briefly, 100% confluent fibroblast were differentiated by adding DMEM/10% FBS/GlutaMAX containing 0.22 $\mu$M dexamethasone, 100 ng/ml biotin, 2 $\mu$g/ml insulin and 500 $\mu$M IBMX (day 0). After 3 d, the medium was replaced by DMEM/10% FBS/GlutaMAX containing 2 $\mu$g/ml insulin. After 3 d (day 6 of differentiation), the medium was replaced with DMEM/10% FBS/GlutaMAX and subsequently replaced every 48 h. Adipocytes were used between day 9 and 12 after the initiation of differentiation. At least 90% of cells were differentiated before experiments.

For GLUT4-translocation, adipocytes between 6 and 7 d after initiation of differentiation were washed twice with PBS, trypsinized, and centrifuged at 200$g$ for 5 min. Supernatant was discarded and the pellet was resuspended in DMEM/10% FBS/2 mM GlutaMAX (13.5 ml for a six wells plate). At this point, cells were mixed with siRNA complexes for knockdown (as described below), or cells were directly seeded into a 96-well plate. Medium was changed 24 h after reseeding (100 $\mu$L per well) and subsequently every 48 h.

SGBS cells were kindly provided by Professor Martin Wabitsch (The University Hospital). Cells were differentiated into adipocytes as previously described (Wabitsch et al, 2001; Fischer-Posovszky et al, 2008) and were used for experiments 12–14 d after differentiation began. Adipocytes were incubated for 4 h in growth-factor-free medium before being treated with insulin.

### Matrigel-coated plates

96-well plates (Eppendorf Cell Imaging plate, UNSPSC 41122107; and PerkinElmer Cell Carrier Ultra, Cat. no. 6055300) were incubated for 2 h at room temperature with a 1:100 matrigel dilution vol/vol in ice-cold PBS. Plates were washed twice with PBS at room temperature before use.

### Assessment of insulin action by immunofluorescent staining of endogenous GLUT4 translocation

Adipocytes were washed twice with warm PBS and serum-starved for 2 h in 50 $\mu$L of DMEM/GlutaMAX/2% BSA with 220 mM bicarbonate (pH 7.4) before stimulation at 37°C with or without insulin (50 $\mu$L/well 2× final insulin concentration) prepared in the same medium. Insulin was made fresh from 750 $\mu$M stock solution on the day of experiment. After stimulation, adipocytes were washed three times with ice cold PBS supplemented with 1 mM calcium and 1 mM magnesium chloride (PBS$^{+/+}$) on ice. Adipocytes were blocked with 10% horse serum or 5% normal swine serum (NSS; Jackson ImmunoResearch 014-000-121) in PBS$^{+/+}$ for 20 min on ice before the addition of test antibodies at 10 $\mu$g/ml in 10% normal goat serum (NGS) or 5% NSS. The exofacial GLUT4 antibodies LM048, LM052 and LM059 were a gift from Joseph Rucker (Integral Molecular). For experiments without TfR staining the primary anti-GLUT4 antibody (1:200) was co-incubated with lectin (L4895-WGA-FITC 488 1:1,000) were incubated for 2 h at 4°C and then cells were washed twice with PBS$^{+/+}$. Cells were subsequently fixed with 4% PFA for 5 min on ice and for 20 min at room temperature. After fixation, cells were washed twice with PBS$^{+/+}$ and incubated with 50 mM glycine in PBS$^{+/+}$ for 5 min at room temperature. Secondary antibody (AF647-conjugated goat anti-human; Jackson Immuno-Research) diluted 1:200 and Hoechst 33342 (1:5,000, H3570; Thermo

Fisher Scientific) in 10% NGS was incubated for 1 h at room temperature. Secondary antibody was removed, cells were washed three times with degassed PBS$^{+/+}$ and 100 $\mu$L/well of degassed imaging buffer was added (2.5% 1,4-diazabicyclo[2.2.2]octane and 10% glycerol, pH 7.8). Solutions with primary or secondary antibodies were filtered through a 0.22 $\mu$m syringe filter (SLGP033NB; Thermo Fisher Scientific) prior used. For intracellular staining, cells were fixed before the blocking step and permeabilized with 10% NGS containing 0.2% saponin. Anti-GLUT4 antibody (rabbit polyclonal antibody generated in-house) was used at 1 $\mu$g/ml and was detected using AF488-conjugated goat anti-rabbit secondary antibody diluted 1:400 (Jackson ImmunoResearch). Note: The LM048 antibody binds to cytoskeletal type structures in permeabilized cells. As a result, fixation-dependent permeabilization can lead to high intracellular staining. For TfR surface labelling, the same protocol was followed, using a monoclonal antibody against TfR (CD71) diluted 1:500 (14-0711-82; Thermo Fisher Scientific) without lectin, and stained with AlexaFluor-conjugated goat anti-Rat secondary antibodies (A48262; Invitrogen).

### Assessment of insulin action using HA-GLUT4-mRuby3

GLUT4 translocation was also measured using overexpressed dual-tagged GLUT4 (HA-GLUT4-mR3). Briefly, the retroviral vector was prepared by using the restriction enzymes XhoI and BamHI to excise Stim1 from pMO91 Stim1 mRuby3 (Nelson et al, 2019). GLUT4 containing HA in the first exofacial loop (Govers et al, 2004) was also excised using XhoI and BamHI and ligated into the pMO91 vector containing mRuby3 to create the HA-GLUT4-mRUBY3 (HA-GLUT4-mR3). 3T3-1 fibroblasts were retrovirally transduced with a construct expressing HA-GLUT4-mR3, differentiated and re-seeded into black 96 well plate as described above. After insulin stimulation, cells were fixed with 4% PFA in PBS for 5 min on ice, then 20 min at room temperature. The PFA was then removed and quenched with 50 mM Glycine in PBS. Cells were then washed once with PBS and then blocked in 5% NSS in PBS for 30 min at room temperature. The quantity of HA at the cell surface was determined by immunofluorescent staining with an anti-HA antibody, consistent with the original HA-GLUT4 reporters published previously (Al-Hasani et al, 1998; Hill et al, 1999; Shewan et al, 2000; Dawson et al, 2001; Govers et al, 2004). The anti-HA antibody (clone 16B12; Covance) was used at 1:1,000 in 2% NSS in PBS, using 45 $\mu$l per well of a 96-well plate, and incubated at room temperature for 1 h. The mRuby3 tag was appended at the C-terminus to avoid interfering with the trafficking of GLUT4 (Dawson et al, 2001), to provide a total GLUT4 signal for surface/total normalization.

### siRNA transfection

siRNA delivery to differentiated adipocytes was performed as previously described (Lundh et al, 2017). Briefly, TransIT-X2 (MIR6006; Mirus) dynamic delivery system was used to transfect adipocytes (6–7 d post differentiation) following the manufacturer's instructions. Optimem/TransIT-X2 was mixed together (ratio 30/1) and kept at room temperature for 20 min. siRNA was added to the optimem/TransIT-X2 to reach a final concentration of 50 nM, mixed gently by pipetting and incubated at room temperature for 30 min.

Co-transfection complex mixture was distributed into 96-well plates. Cells were reseeded on top of the transfection reagents as it was described above. After 24 h, the medium was replaced with DMEM/10% FBS/GlutaMAX. Re-seeding as part of the reverse transfection protocol maximizes gene knockdown (data not shown) and also minimizes any differences between control and knockdown cells that are due to well-to-well variation in differentiation efficiency.

### In vitro induction of insulin resistance

Long-term Insulin resistance models were induced by chronic stimulation with dexamethasone (dex), TNF-α or hyperinsulinemia as previously described (Fazakerley et al, 2018). Chronic inflammation was generated by incubation of 2 ng/ml of TNF (calbiochem) during 4 d. The medium was replaced every 24 h. To induce hyperinsulinemia, cells were incubated with 10 nM insulin at 1,200, 1,600 and 2,000 h on day 1 and 0800 h the following day (Hoehn et al, 2008; Fazakerley et al, 2018). Glucocorticoid-induced insulin resistance was recreated with 20 nM dexamethasone (Dex, 0.01% ethanol carrier as control), and the medium was replaced every 48 h for 8 d. Short term treatment consisted in 24 h stimulation with a single administration of TNF or insulin, respectively.

### Western blotting

After serum starvation or siRNA knockdown (as described above), the cells were placed on ice, washed with ice cold PBS, lysate with 1% (wt/vol) SDS in PBS or radioimmunoprecipitation assay (RIPA) buffer containing protease inhibitors (Roche Applied Science), and tip-sonicated (90% for 10 s). Lysates were centrifuged at 13,000$g$ for 15 min at 4°C. The lipid layer was removed by aspiration and protein content was quantified by PierceTM BCA protein assay kit (Thermo Fisher Scientific). 10 $\mu$g of lysate was resolved by SDS–PAGE, transferred to PVDF membranes, and immunoblotted as described previously (Norris et al, 2017). Primary antibodies used in this study include IRS-1 (CST, Cat. no. 3407), *Pan* Akt (CST, Cat. no. 2920), AS160/TBC1D4 (CST, Cat. no. 2670), Rab10 (Invitrogen, Cat. no. MA515670). In-house anti-GLUT4, α-Tubulin (Sigma-Aldrich, Cat. no. T9026), Gapdh (CST, Cat. no. 2118) and 14-3-3 (Santa Cruz) were used as house-keeping controls. Image Studio Lite Ver 5.2 (LI-COR) or ImageJ were used to perform the densitometry analysis. Statistical tests were performed using GraphPad Prism version 9.0.

### Sample preparation and real-time quantitative-PCR assays

For Fig S3D, total RNA was extracted from cells using QIAshredder and RNeasy kits (QIAGEN). To remove any DNA, the extracts were incubated with DNAse buffer (Promega) and residual DNAse was subsequently inactivated with DNAse stop solution (Promega). cDNA synthesis was performed using LunaScript RT SuperMix Kit (NEB). Polymerase chain reactions were carried out using TaqMan 2X Universal PCR Master Mix or SYBR Green PCR Master Mix (Thermo Fisher Scientific) on a QuantStudio 7 Flex Real-Time PCR System (Thermo Fisher Scientific). Acidic ribosomal phosphoprotein P0 (36B4), β-actin (b-act), and 18S ribosomal RNA (18 s) were used as internal controls. The following primer sets were used: 36B4_F;

AGATGCAGCAGATCCGCAT and 36B4_R; GTTCTTGCCCATCAGCACC, b-act_F; GCTCTGGCTCCTAGCACCAT and b-act_R; GCCACCGATC-CACACAGAGT, and 18s_F; CGGCTACCACATCCAAGGAA and 18s_R; GCTGGAATTACCGCGGCT, with the corresponding 18s TaqMan probe GAGGGCAAGTCTGGTGCCAG. The TaqMan gene expression assay (premixed primer set and probes) was used for mouse *Bcl9l* (Mm01143422_m1).

### Glucose uptake

After 2-h serum starvation, glucose transporter activity was measured as previously described (Fazakerley et al, 2015). Briefly, after treatment, the cells were washed three times with PBS, incubated with Krebs-ringer 0.2% BSA buffer (KRP buffer; 0.6 mM $Na_2HPO_4$, 0.4 mM $NaH_2PO_4$, 120 mM NaCl, 6 mM KCl, 1 mM $CaCl_2$, 1.2 mM MgSO4, and 12.5 mM Hepes [pH 7.4]) without any drugs. Cells were stimulated with 1 nM insulin for 20 min. No-specific glucose incorporation was determined by adding 25 $\mu$M of cytochalasin B (ethanol; Sigma-Aldrich) before 2-[3H]deoxyglucose (2-DOG) (PerkinElmer). 2-DOG was added during the last 5 min of stimulation (0.25 $\mu$Ci, 50 $\mu$M). Then, cells were washed three times with ice-cold PBS and permeabilized with Triton X-100 (1% vol/vol in PBS). 2-DOG was determined by liquid scintillation counting and data normalized against protein concentration. Data were further normalized to basal control cells.

### Microscope image acquisition and analysis

We used PerkinElmer's Opera Phenix TM high-throughput and high-content imaging system acquisition in a 96-well glass bottom plate (Eppendorf Cell Imaging plate, UNSPSC 41122107; and PerkinElmer Cell Carrier Ultra, Cat. no. 6055300) with a 20× NA1.0 water immersion objective in confocal mode. Nine fields of view were taken by well. Basal surface analysis was performed using an automated pipeline developed in Harmony 4.9, a high-content image acquisition and analysis system provided by PerkinElmer. Rim analysis was performed using a semi-automated approach. Individual channels were merged into multichannel tiffs using Fiji (Schindelin et al, 2012) and bioformats. Segmentation of cell nuclei, PM rims, cytoplasm and lipid droplets was performed using a supervized machine learning approach implemented in Ilastik (Berg et al, 2019) (www.ilastik.org). The classifier was developed by training on 20–25 images per experiment. The resulting prediction maps and the primary images were then analysed using a macro written in Fiji (Schindelin et al, 2012) Data were collated using Microsoft Power Query for Excel and processed using Tableau prep builder 2021.1. All analysis scripts and pipelines are available on request.

#### *Optimisation of image acquisition and analysis*

Image analysis using an analysis pipeline developed in PE Harmony has several drawbacks when considering a screening approach. For instance, it requires flat plates and well-aligned objectives to capture the basal surface, the basal membrane delivers a relatively weak signal that is more susceptible to noise, and the analytical burden of single cell data is large and the software unsuited to large scale screening. As such, we tested quantification of surface levels of GLUT4 at different planes of the surface membrane. The use of a plane located in the central part of the cell (rims) provided

greater signal intensity and sensitivity, facilitated optimal distinction between adipocytes and fibroblasts (using the lipid droplet content), yielded increased imaging speeds, and appeared less sensitive to plate to plate variance.

### Single cell analysis

Adipocytes were distinguished from fibroblasts and necrotic cells in the cell cultures based on light scatter from the lipid droplets (Muretta et al, 2008). We describe the $i$th plate of cells as a 1xN vector $X_i$, where N refers to the number of cells measured. To normalize plate to each other, we first determined the median values of control 0 nM insulin and control 100 nM insulin cells for each plate, termed $med0_i$ and $med100_i$. The maximum values of these two groups across plates were computed as $maxmed0 = max\{med0_i\}$ and $maxmed100 = max\{med100_i\}$. Then, each plate was linearly transformed to make the control 0 nM insulin medians equal to $maxmed0$ and control 100 nM insulin medians equal to $maxmed100$ that is:

$$Xnorm_i = \frac{(X_i - med0_i) * (maxmed100 - maxmed0)}{(med100_i - med0_i)} + maxmed0.$$

Finally, the $\log_2$ of all normalized values was taken and used for subsequent analysis. Density plots of single cell data were generated using the r package ggridges.

### Sample preparation for TRARG1 interactome analysis

Serum-starved (basal) or insulin-stimulated (100 nM insulin, 20 min) 3T3-L1 adipocytes were washed thrice with ice-cold PBS and lysed in lysis buffer (1% [wt/vol] CHAPS, 150 mM NaCl, 50 mM Tris–HCl, pH 7.5, and 5% [vol/vol] glycerol with protease inhibitor mixture and phosphatase inhibitors) by passing through a 22-gauge needle six times, followed by a 27-gauge needle three times. Octyl $\beta$-D-glucopyranoside was added into each lysate to a final concentration of 50 mM and lysates were solubilized for 30 min with rotation at 4°C before centrifugation at 20,000$g$ for 20 min at 4°C to remove lipid and cell debris. The resulting supernatants were transferred into a new tube and centrifuged at 265,100$g$ for 30 min. 8 mg of each lysate was incubated with 16 $\mu$l of anti-TRARG1 antibody (Santa Cruz: sc-377025) or the same amount of IgG of the same species for 1 h with rotation at 4°C. 100 $\mu$l Magnetic Dynabeads were added into each antibody-lysate mixture and incubated for 1 h with rotation at 4°C. Beads were washed three times with 150 mM NaCl, 50 mM Tris–HCl, pH 7.5, 5% glycerol, 0.01% CHAPS, and twice with 150 mM NaCl, 50 mM Tris–HCl, pH 7.5, 5% glycerol. For MS sample preparation, beads were incubated in 25 $\mu$l 2 M urea, 50 mM Tris–HCl, pH 7.5, 5 mM TCEP, 20 mM 2-chloroacetamide, and 5 $\mu$g/ml trypsin for 30 min at room temperature, before 100 $\mu$l 2 M urea, 50 mM Tris–HCl, pH 7.5, was added. Each eluate was collected into a LowBind Eppendorf tube and digested overnight at RT. Peptides were acidified by adding TFA to a final concentration of 1%.

StageTips containing 2X SDB-RPS (Empore, 3M) were pre-equilibrated with addition of 100% ACN, 30% methanol (Thermo Fisher Scientific) and 1% TFA then 0.2% TFA and 5% ACN and centrifugation at 1,500$g$ for 3 min. Peptide samples were loaded and washed with 1% TFA in isopropanol, then 0.2% TFA and 5% ACN.

Peptides were eluted from the SDB-RPS tips in 60% ACN and 20% $NH_4OH$ (25%, HPLC grade; Sigma-Aldrich). Samples were dried in a vacuum concentrator for 50 min at 45°C and resuspended in 2% ACN/0.3% TFA for LC–MS/MS analysis.

### Mass spectrometry

Mass spectrometry analysis was performed on a LC/MS system including an EASY-nLC coupled to a Q Exactive HF-X mass spectrometer. Peptides were separated using a 75-$\mu$m × 40-cm column packed in-house (ReproSil Pur C18-AQ, 1.9 $\mu$m particle size) with a gradient of 5–40% (EASY-nLC) buffer B (80% ACN/0.1% FA) over 40 min at 300 nl/min with the column maintained at 60°C using a column oven. The mass spectrometer was operated in data-dependent acquisition mode, acquiring survey scans of $3 \times 10^6$ ions at a resolution of 60,000 from 300 to 1,650 m/z. 20 of the most abundant precursors from the survey scan with charge state >1 and <6 were selected for fragmentation. Precursors were isolated with a window of 1.4 m/z and fragmented in the HCD cell with NCE of 27. Maximum ion fill times for the MS/MS scans were 28 ms and target fill value was $1 \times 10^4$ ions. Fragment ions were analysed with high resolution (15,000) in the Orbitrap mass analyser. Dynamic exclusion was enabled with duration 15–20 s.

### Processing of spectral data and data analysis

Raw mass spectrometry data were processed using the Andromeda algorithm integrated into MaxQuant (v1.6.6.0 or v1.6.1.0) (Cox & Mann, 2008), searching against the mouse UniProt database (June 2019 release) concatenated with known contaminants. Default peptide modification settings were used and match between runs was turned on with a match time window of 0.7 min and an alignment time window of 20 min for identifications transfer between adjacent fractions. Protein and peptide FDRs were filtered to 1%, respectively.

This study was performed with three biological replicates. The values for each IgG replicate were generated by averaging each IgG replicate under the two conditions (basal, insulin) to eliminate variances between conditions driven by IgG control. Proteins were filtered out for reverse sequences, potential contaminants, peptides only identified by site and proteins with less than two quantified values in all bait conditions. LFQ intensities were $\log_2$-transformed and median normalized. Missing values were imputed as previously described (Yang et al, 2019). Briefly, if values were quantified in at least two replicates for a condition, the remaining missing values would be imputed by sampling from a distribution with the same mean and SD as the quantified replicates for that particular condition and protein. For remaining missing values in IgG controls, a second step of imputation was performed using a downshifting approach where missing values were replaced by sampling from a normal distribution that resembles the low tail of the distribution of quantified values in each sample (down shifted mean = sample mean - 1× sample SD, down shifted SD = 0.6× sample SD) (Robles et al, 2017). To characterise the changes in protein abundance in bait condition compared with IgG control condition, $\log_2$ fold changes were estimated and two sample $t$ tests performed for the proteins with no missing values after imputation. $P$-values were corrected for multiple hypothesis testing using the Benjamini

and Hochberg method (Benjamini & Hochberg, 1995). To determine which proteins are differently enriched in response to insulin, median absolute deviation (MAD) was calculated based on the variability of $\log_2$ fold change (Bait-IgG) of all quantified proteins between insulin and basal conditions. Proteins that have an absolute $\log_2$ fold-change (Bait-IgG) difference greater than 2*MAD between basal and insulin conditions, a $\log_2$ fold-change (Bait-IgG) greater than 2, and adj. $P \leq 0.05$ are labelled as insulin-regulated.

### Subcellular fractionation

Day 10 HA-GLUT4-mR3 adipocytes were subjected to differential centrifugation subfractionation as previously described (Fazakerley et al, 2015). Briefly, cells were washed with ice-cold PBS and harvested in ice-cold HES-I buffer (20 mM HEPES, pH 7.4, 1 mM EDTA, 250 mM sucrose containing Complete protease inhibitor mixture [Roche Applied Science]). All subsequent steps were carried out at 4°C. Cells were homogenized using a Dounce homgenizer before centrifugation at 500$g$ for 10 min. The supernatant was centrifuged at 13,550$g$ for 12 min to pellet the PM and mitochondria/nuclei. The supernatant was then centrifuged at 21,170$g$ for 17 min to pellet the high density microsomal (HDM) fraction. The supernatant was again centrifuged at 235,200$g$ for 75 min to obtain the cytosol fraction (supernatant) and the low density microsomal (LDM) fraction (pellet). The PM and mitochondria/nuclei pellet were resuspended in HES-I and layered over a high-sucrose HES-I buffer (1.12 M sucrose, 0.05 mM EDTA, and 10 mM HEPES, pH 7.4) and centrifuged at 111,160$g$ for 60 min in a swing-out rotor. The PM fraction was collected above the sucrose layer, and the pellet was the mitochondria/nuclei fraction. All the fractions were resuspended in HES-I. Protein concentration for each fraction was performed using BCA assay (Thermo Fisher Scientific). All lysates were then resolved as per the Western blotting protocol above and labelled with the primary antibodies Caveolin-1 (CST, Cat. no. 3267) and the in-house anti-GLUT4 antibody.

### Statistical analysis

GraphPad Prism 9 (GraphPad Software Inc.) was used for statistical analyses. If not otherwise stated, data are expressed as the mean ± SD. Relevant statistical tests are noted in the figure legends.

## Data Availability

Raw and MaxQuant processed data of mass spectrometry data for the TRARG1 interactome have been deposited in the PRIDE ProteomeXchange Consortium (http://proteomecentral.proteomexchange.org/cgi/GetDataset) (Perez-Riverol et al, 2019) and can be accessed with the identifier PXD022765.

## Supplementary Information

## Acknowledgements

This work was supported by National Health and Medical Research Council (NHMRC) Project Grant GNT1120201 and GNT1061122 (to DE James). DE James is an Australian Research Council (ARC) Laureate Fellow. DJ Fazakerley was supported by a Medical Research Council Career Development Award (MR/S007091/1) and a Wellcome Institution Strategic Support Fund award (204845/Z/16/Z). JG Burchfield and A Diaz-Vegas were supported by NHMRC Ideas Grant (GNT2013621), the Diabetes Australia Research Program (Y22G-DIAA), and the Mitochondrial Foundation (Mitofoundation, G057). The content is solely the responsibility of the authors and does not necessarily represent the official views of the NHMRC or ARC. The authors also acknowledge the facilities, and the scientific and technical assistance of Sydney Cytometry and the Sydney Mass Spectrometry Facility, at the Charles Perkins Centre, University of Sydney. These studies were supported by the Wellcome-MRC, Institute of Metabolic Science, Metabolic Research Laboratories, Imaging Core (Wellcome Trust Major Award [208363/Z/17/Z]). We also acknowledge and thank the image.sc forum (https://forum.image.sc/) for their help and support in developing the image analysis pipelines.

### Author Contributions

A Diaz-Vegas: conceptualization, data curation, formal analysis, funding acquisition, investigation, methodology, and writing—original draft, review, and editing.
DM Norris: conceptualization, formal analysis, validation, investigation, visualization, methodology, and writing—original draft, review, and editing.
S Jall-Rogg: formal analysis, investigation, visualization, and methodology.
KC Cooke: data curation and investigation.
OJ Conway: formal analysis and investigation.
AS Shun-Shion: formal analysis and investigation.
X Duan: formal analysis and investigation.
M Potter: investigation.
J van Gerwen: formal analysis, investigation, visualization, and writing—original draft.
HJM Baird: formal analysis and investigation.
SJ Humphrey: formal analysis and investigation.
DE James: conceptualization, supervision, funding acquisition, writing—original draft, and project administration.
DJ Fazakerley: conceptualization, data curation, formal analysis, supervision, funding acquisition, validation, investigation, methodology, project administration, and writing—original draft, review, and editing.
JG Burchfield: conceptualization, data curation, formal analysis, supervision, funding acquisition, validation, investigation, methodology, project administration, and writing—original draft, review, and editing.

### Conflict of Interest Statement

The authors declare that they have no conflict of interest.

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
