## [Reviewer comments · Life Science Alliance]

Life Science Alliance

A high-content endogenous GLUT4 trafficking assay reveals new aspects of adipocyte biology

Alexis Diaz-Vegas, Dougall Norris, Sigrid Jall-Rogg, Kristen Cooke, Olivia Conway, Amber Shun-Shion, Xiaowen Duan, Meg Potter, Julian van Gerwen, Harry Baird, Sean Humphrey, David James, Daniel Fazakerley, and James Burchfield

DOI: <https://doi.org/10.26508/lsa.202201585>

Corresponding author(s): James Burchfield, University of Sydney and David James, School of Life and Environmental Sciences, the University of Sydney

Review Timeline:

Submission Date:	2022-06-29
Editorial Decision:	2022-06-30
Revision Received:	2022-09-23
Editorial Decision:	2022-09-23
Revision Received:	2022-10-10
Accepted:	2022-10-10

Transaction Report:

Please note that the manuscript was previously reviewed at another journal and the reports were taken into account in the decision-making process at Life Science Alliance.

Referee #1 Review

Report for Author:

This study provides a robust image-based platform to understand the regulation of endogenous Glut4 trafficking in cultured adipocytes. This method validated the role of previously reported regulators of Glut4 as well as identified Kif13A as a novel regulator of Glut4 traffic. The study is a tour de force of Glut4 regulation - providing a powerful tool to the scientific community. The study is highly rigorous and commendable.

I only have minor comments/suggestions:

1. The study was performed in immortalized adipocytes, such as 3T3-L1 cells. This cell line has been widely used in the field of fat biology and is justifiable. Meanwhile, this platform also provides a terrific opportunity to study the difference in Glut4 trafficking in visceral vs. subcutaneous-derived adipocytes using primary adipocytes. The authors wish to explore this if resources are available.
2. The authors described that this system applies to adipocytes but not to myocytes. The title and abstract should clarify that this system works in cultured adipocytes.

Referee #2 Review

Report for Author:

In this manuscript, the authors describe a high-content imaging platform for studying translocation to the plasma membrane of endogenous glucose transporter GLUT4. It is based on recently developed antibodies that recognize the extracellular domain of GLUT4 (Tucker et al, 2018). They first showed that the antibodies can detect a specific signal at the surface of 3T3-L1 and SGBS adipocytes following stimulation with insulin. They then validated the approach (translocation of endogenous GLUT4) for studying insulin resistance in classical cell culture models. The second part of the manuscript describes the use of the platform for the identification of regulators of GLUT4 trafficking. These experiments were implemented by the simultaneous analysis of transferrin receptor (TfR) trafficking, nuclei counting and visualization of lipid droplets. In addition of known regulators of GLUT4

trafficking such as RAB10, this analysis allowed the identification of the motor protein KIF3A as a new regulator and of Bcl9l, which is involved in both GLUT4 and TfR trafficking. Finally, the authors compared cell surface expression of endogenous GLUT4 and TfR in wild-type 3T3-L1 and cells expressing HA-GLUT4-mRuby3 following knockdown of a series of genes. Although it provides some new information on the genes involved in GLUT4 trafficking, this manuscript is essentially methodological. I am therefore not sure that it fully corresponds to the aims and scope of this journal. Having said that, the experiments are well done and generally convincing. The analytical tools developed in this study will certainly benefit the GLUT4 field.

Main concern:

- The main difficulty encountered in the GLUT4 field is that most studies have been performed in cells expressing exogenous tagged GLUT4 constructs. These cellular models have sometimes led to contradictory results, linked to overexpression problems or modifications of intracellular trafficking induced by the tag. Following the traffic of the endogenous receptor is therefore a great advantage. The authors have addressed this point in the experiments shown in Figure 7. It should however be more developed, which in my opinion would increase the impact of this paper. First, the Figure legend is quite unclear. I guess that the color codes correspond to the ones shown in Fig. 6. But what about the grey circles? I agree that the slopes of the curves argue against differences in knock-down efficiency between the two cell lines. However, this could be verified at least for Akt1/2 knock-down. Does pharmacological inhibition of Akt lead to the same effect in HA-GLUT4-mRuby3 cells? The experiments have been performed after stimulation with 0.5 nM insulin. It would be important to perform these experiments with no stimulation and 100 nM insulin as in Fig. 6. Finally, it would be informative to document the level of overexpression of the HA-GLUT4-mRuby3 compared to endogenous GLUT4.

Minor points:

- The LM052 and LM059 antibodies are conformation insensitive whereas LM048 is state-specific (Tucker et al, 2018). Most of the experiments have been performed with the LM048 antibody. Does it make a difference? The authors could comment on it.
- Fig. 1B: a western blot showing the efficiency of GLUT4 depletion should be provided.
- Fig. 1C, right: I guess that the legend of the ordinate should be total GLUT4 instead of pmGLUT4?
- Fig. 2B: the legend is unclear. What are the three tracks for each condition?
- Discussion (1st page): Fig. 8 does not exist.

Referee #3 Review

Report for Author:

Diaz-Vegas and colleagues study endogenous Glut4 trafficking/localization using an antibody directed against the luminal epitope of the protein. Thanks to this Ab the authors were able to follow the insulin-induced redistribution of endogenous Glut4 to the plasma membrane in non-permeabilized cells and its modulation by insulin signalling inhibitors or by conditions mimicking insulin resistance. They generally found that the endogenous Glut4 trafficking was more sensitive than that of the overexpressed tagged one (the only Glut4 form so-far characterized) to insulin regulation.

They then exploited the anti-luminal Ab to establish a platform for high content screening by running a pilot screen with siRNA for known regulators of insulin-stimulated trafficking of overexpressed tagged-GLUT4, assessing at the same time endogenous Glut4 and TfR exposure at the plasma membrane. They found that the majority of known GLUT4 regulators altered insulin-induced plasma membrane exposure of endogenous GLUT4. Of note, they found that Kif3 also impairs endogenous Glut4 trafficking, confirming previous results involving kinesins in Glut4 trafficking (PMID: 22473005, PMID: 12832475).

The authors also found some exceptions: Akt1/2, Lnpep and Cltc depletion, which impair the plasma membrane exposure of tagged Glut4, did not affect the plasma membrane exposure of the endogenous Glut4. Rab14 depletion, which is known to decrease the plasma membrane exposure of tagged Glut4, instead increased that of the endogenous Glut4, while Tbc1d4/As160 depletion that increases the insulin-induced trafficking of tagged-GLUT4 impaired that of the endogenous Glut4. In a few cases the authors explain the discrepancy as being due to the ability of the regulators to affect not only the trafficking but also the expression of the endogenous Glut4 (the expression from the plasmid encoding the tagged form being insensitive to this kind of regulation). Unfortunately, the authors do not "exploit" any of the discrepancies they uncovered to assess whether and how the trafficking of the endogenous Glut4 has different molecular requirements as compared to the overexpressed one. Thus, while in general the data are convincing and set the basis for future studies on the regulation of the endogenous Glut4 distribution, they do not lead to a significant advance in our mechanistic understanding of Glut4 trafficking.

June 30, 2022

Re: Life Science Alliance manuscript #LSA-2022-01585-T

Dr. James G Burchfield
The University of Sydney
The Charles Perkins Centre
The University of Sydney
Sydney, NSW 2006
Australia

Dear Dr. Burchfield,

Thank you for submitting your manuscript entitled "A high-content endogenous GLUT4 trafficking assay reveals new aspects of GLUT4 biology" to Life Science Alliance. We invite you to submit a revised manuscript addressing the following points:

- Address Reviewer 1's point #2
- Address Reviewer 2's comments

Thank you for this interesting contribution to Life Science Alliance. We are looking forward to receiving your revised manuscript.

Sincerely,

- A letter addressing the reviewers' comments point by point.
- An editable version of the final text (.DOC or .DOCX) is needed for copyediting (no PDFs).
- High-resolution figure, supplementary figure and video files uploaded as individual files: See our detailed guidelines for preparing your production-ready images, <https://www.life-science-alliance.org/authors>
- Summary blurb (enter in submission system): A short text summarizing in a single sentence the study (max. 200 characters including spaces). This text is used in conjunction with the titles of papers, hence should be informative and complementary to the title and running title. It should describe the context and significance of the findings for a general readership; it should be written in the present tense and refer to the work in the third person. Author names should not be mentioned.
- By submitting a revision, you attest that you are aware of our payment policies found here: <https://www.life-science-alliance.org/copyright-license-fee>

B. MANUSCRIPT ORGANIZATION AND FORMATTING:

Reviewer comments have been bulleted and *italicized* for clarity.

Reviewer 1

- *2. The authors described that this system applies to adipocytes but not to myocytes. The title and abstract should clarify that this system works in cultured adipocytes.*

Response: To address the reviewers concerns, we have changed the title to “A high-content endogenous GLUT4 trafficking assay reveals new aspects of GLUT4 biology **in cultured adipocytes**” and clarified the abstract as follows “Here we describe a high-content imaging platform for studying endogenous GLUT4 translocation **in intact adipocytes**”

Reviewer 2

- *The main difficulty encountered in the GLUT4 field is that most studies have been performed in cells expressing exogenous tagged GLUT4 constructs. These cellular models have sometimes led to contradictory results, linked to overexpression problems or modifications of intracellular trafficking induced by the tag. Following the traffic of the endogenous receptor is therefore a great advantage.*
- *The authors have addressed this point in the experiments shown in Figure 7. It should however be more developed, which in my opinion would increase the impact of this paper*
- *First, the Figure legend is quite unclear. I guess that the color codes correspond to the ones shown in Fig. 6. But what about the grey circles?*

Response: A legend with the identity of all the points in the figure has been included and all the points are now coloured. The figure legend now reads as follows:

Figure 7. Trafficking of overexpressed HA-GLUT4-mR3 protects cells from genetic perturbations in comparison to endogenous GLUT4. (A), (C) and (E) - correlations between the endogenous GLUT4 or HA-GLUT4-mR3 trafficking responses in wild-type and HA-GLUT4-mRuby3-expressing cells in response to siRNA knockdown of several regulators of GLUT4 trafficking (as indicated). **(B), (D) and (F)** - correlations between the TfR trafficking responses in wild-type and HA-GLUT4-mRuby3-expressing cells in response to siRNA knockdown of regulators of GLUT4 trafficking (as indicated). Plasma membrane GLUT4 (pmG4) and TfR were determined under basal **(A and B)**, 0.5 nM insulin **(C and D)** or 100 nM insulin-stimulated conditions **(E and F)**. FOC = Fold over control. Correlations are indicated by the respective r^2 and p-values as determined by simple linear regression.

- *I agree that the slopes of the curves argue against differences in knock-down efficiency between the two cell lines. However, this could be verified at least for Akt1/2 knock-down.*

Response: To address this, we assessed the knockdown of *Akt1/2* and *Rab10* in both cell lines (new Fig. S5). We observed no significant difference in the knockdown efficiency between the cell lines and have adjusted the text with the following:

In the results:

“The distinct GLUT4 and HA-GLUT4-mR3 responses to gene knockdown are unlikely explained by differences in knockdown efficiency between cell lines, given that the effects on TfR are so consistent and that no difference was observed in the depletion of Akt1/2 and Rab10 between the cell lines (Fig S4). Therefore, these data suggest that GLUT4 overexpression has a specific effect on the sensitivity of GLUT4 trafficking responses to gene knockdown.”

In the discussion:

“Further, adipocytes overexpressing GLUT4 were less sensitive to knockdown of known GLUT4 regulators despite near identical results for TfR in the same cells (Fig. 7) and no significant differences in knockdown of tested targets between the cell lines (Fig. S4)”

- *Does pharmacological inhibition of Akt lead to the same effect in HA-GLUT4-mRuby3 cells?*

Response: We have performed an Akt inhibitor (MK2206) dose response study in both cell lines to test the sensitivity of endogenous or HA-GLUT4-mR3 to Akt inhibition (Fig S5). No difference in sensitivity to MK2206 was observed in insulin stimulated GLUT4 translocation. In contrast, we observed a significant decrease in sensitivity to MK2206 with GLUT4 overexpression in insulin stimulated TfR translocation. It is important to note that the overall sensitivity to MK2206 for the TfR response is markedly right shifted, consistent with a greater insulin sensitivity of this response. This suggests that overexpression of GLUT4 likely has a modest effect on overall insulin sensitivity.

We have amended the results as follows:

This is interesting given that Akt is a central node in the insulin signalling pathway and these data may indicate change in insulin sensitivity or network rewiring as a result of GLUT4 overexpression. To further explore this, we tested the sensitivity of both WT and HA-GLUT4-mR3 over-expressing lines to the Akt inhibitor MK2206 at submaximal (1 nM) and maximal (100 nM) insulin. No difference in sensitivity to MK2206 was observed in insulin-induced GLUT4 translocation between WT and HA-GLUT4-mR3 cells (Fig. S5C,D), suggesting that the differences in response to *Akt1/2* knockdown between WT and GLUT4 over-expressing may be due to altered responses to prolonged lower expression of Akt isoforms, rather than the sensitivity of GLUT4 to Akt activity, or differences in KD efficiency when these experiments were performed.

- *The experiments have been performed after stimulation with 0.5 nM insulin. It would be important to perform these experiments with no stimulation and 100 nM insulin as in Fig. 6.*

Response: Figure 7 now contains the comparison at basal and 100nM in addition to the 0.5nM insulin stimulation. The manuscript has been amended to reflect this as follows.

In the results:

“We next compared the responses of endogenous GLUT4 and HA-GLUT4-mR3 to gene depletion. We performed the same series of knockdowns as in Figure 6 in adipocytes expressing HA-GLUT4-mR3, and correlated surface GLUT4 and TfR levels in each cell line under basal or insulin-stimulated conditions across all knockdowns (0.5 nM and 100 nM insulin).

Under basal conditions there was little effect of knockdown on surface GLUT4 levels in either cell line and unsurprisingly, there was no correlation between them ($r^2=0.01$, slope=-0.17) (Fig. 7A). In response to either submaximal or maximal insulin concentrations, the impact of knockdown on surface GLUT4 levels in HA-GLUT4-mR3 cells was weaker compared to that of endogenous GLUT4, as demonstrated by the relatively flat slopes of the correlation (slope = 0.16 for both submaximal and maximal insulin) and the low correlation coefficient ($r^2=0.08$ for both submaximal and maximal insulin) (Fig. 7C & E). In contrast, the correlation between cell lines was substantially higher for surface TfR under basal conditions ($r^2=0.3$, $p<0.0001$), and upon insulin stimulation ($r^2=0.45$, $p<0.0001$ for submaximal and $r^2=0.47$, $p<0.0001$ for maximal insulin) with a broadly equivalent effect size (Slope = 1.02 and 0.88 for submaximal and maximal insulin concentration, respectively) (Fig. 7D & F).”

The corresponding figure legend is as follows:

Figure 7. Trafficking of overexpressed HA-GLUT4 protects cells from genetic perturbations in comparison to endogenous GLUT4. (A), (C) and (E) - correlations between the endogenous GLUT4 or HA-GLUT4 trafficking responses in wild-type and HA-GLUT4-mRuby3-expressing cells, respectively, in response to siRNA knockdown of several regulators of GLUT4 trafficking. **(B), (D) and (F)** - correlations between the TfR trafficking responses in wild-type and HA-GLUT4-mRuby3-expressing cells, in response to siRNA knockdown of regulators of GLUT4 trafficking (as indicated). Plasma membrane GLUT4 (pmG4) and TfR were determined under basal (A and B), and 0.5 nM (C and D) or 100 nM insulin-stimulated conditions (E and F). FOC = Fold over control. Correlations are indicated by the respective r^2 and p-values as determined by simple linear regression.

- *Finally, it would be informative to document the level of overexpression of the HA-GLUT4-mRuby3 compared to endogenous GLUT4.*

Response: Based on maximal surface labelling with the LM048 antibody (n=6) and subcellular fractionation data (n=1), we conclude that the HA-G4-mR cells have ~2 times the GLUT4 concentration per cell as the endogenous cell line.

Results:

We next compared the responses of endogenous GLUT4 and overexpressed HA-GLUT4-mRuby3 (HA-GLUT4-mR3) to gene depletion. Western blotting revealed an overexpression of HA-GLUT4-mR3 that was similar to the level endogenous GLUT4, indicating that these cells contain roughly twice the amount of total GLUT4 as wild-type cells (Fig. S5A). Subcellular fractionation revealed a highly similar distribution of HA-GLUT4-mR3 to endogenous GLUT4 under basal and insulin stimulated conditions (Fig. S5A). This was supported by surface labelling of GLUT4 in response to 100 nM insulin (using the LM048 antibody), whereby the PM GLUT4 signal was ~2 times greater in HA-GLUT4-mR3 cells compared with WT cells (Fig. S5B)

In the legend:

Figure S5. Western blotting of cell lysates for GLUT4 and Caveolin1 (CAV1) following subcellular fractionation in the presence and absence of 100 nM insulin for 30 minutes. WCH - Whole cell homogenate; M/N - mitochondrial/nuclear; HDM - high density microsomes; LDM low density microsomes; PM - plasma membrane. n=1. **(A).** PM GLUT4 levels in response to 100 nM insulin in wild type and HA-GLUT4-mR3 overexpressing cells n = 2 **(B).** PM GLUT4 abundance in wild-type or HA-G4 overexpressing 3T3-L1 adipocytes treated with 1 nM insulin **(C)** or 100 nM insulin **(D)** in combination with DMSO (control) or the Akt inhibitor MK2206 for 10 minutes prior insulin addition. Sensitivity to MK2206 in WT cells in response to 1 nM or 100 nM insulin for GLUT4 **(E)** is shown. Data was min:max normalised for each insulin concentration. n = 4. Non-linear regression comparing independent fits with a global fit shared parameter (IC50) was used to test statistical significance.

In the Method:

Subcellular fractionation:

Day 10 HA-GLUT4-mR3 adipocytes were subjected to differential centrifugation subfractionation as previously described (Fazakerley et al, 2015a). Briefly, cells were washed with ice-cold PBS and harvested in ice-cold HES-I buffer (20 mM HEPES, pH 7.4, 1 mM EDTA, 250 mM sucrose containing Complete protease inhibitor mixture (Roche Applied Science)). All subsequent steps were carried out at 4 °C. Cells were homogenized using a dounce homogeniser prior to centrifugation at 500 g for 10 min. The supernatant was centrifuged at 13,550 g for 12 min to pellet the PM and mitochondria/nuclei. The supernatant was then centrifuged at 21,170g for 17 min to pellet the high density microsomal (HDM) fraction. The supernatant was again centrifuged at 235,200 g for 75 min to obtain the cytosol fraction (supernatant) and the low density microsomal (LDM) fraction (pellet). The PM and mitochondria/nuclei pellet were resuspended in HES-I and layered over a high sucrose HES-I buffer (1.12 M sucrose, 0.05 mM EDTA, 10 mM HEPES, pH 7.4) and centrifuged at 111,160 g for 60 min in a swing-out rotor. The PM fraction was collected above the sucrose layer, and the pellet was the mitochondria/nuclei fraction. All the fractions were resuspended in HES-I. Protein concentration for each fraction was performed using BCA assay (Thermo Scientific). All lysates were then resolved as per the western blotting protocol above and labelled with the primary antibodies Caveolin-1 (CST, Cat# 3267) and the GLUT4 1F8 Antibody.

Minor points:

- The LM052 and LM059 antibodies are conformation insensitive whereas LM048 is state-specific (Tucker et al, 2018). Most of the experiments have been performed with the LM048 antibody. Does it make a difference? The authors could comment on it.

Response: We have added the following to the Discussion.

The LM048 antibody was reported to be state-dependent for human GLUT4, favouring the outward confirmation (Tucker et al, 2018). Although we have not tested this directly, we performed all fixation or antibody staining in assays using this antibody in the absence of glucose, which would likely promote this outward confirmation and favour antibody binding. However, one limitation of using this LM048 antibody is that experimental interventions that alter GLUT4 conformation might affect antibody binding and therefore falsely report changes in PM GLUT4. Nevertheless, these antibodies represent an advance for studying endogenous GLUT4 trafficking and may offer means to study PM appearance of endogenous GLUT4 in vivo or ex vivo, including muscle tissues.

Fig. 1B: a western blot showing the efficiency of GLUT4 depletion should be provided.

Response: We have provided images for the siRNA knockdown of GLUT4 by immunofluorescence (IF) with two different antibodies, for surface and total GLUT4 (Fig. 1B) and the quantification of these data (Fig. 1C). We recognised a typo in the right panel of Fig. 1C, which is meant to say “Total GLUT4” and have rectified this. It is possible that this typo has led the reviewer to believe there was no total GLUT4 quantification. Given the effect observed for total GLUT4 knockdown by IF, we see no additional benefit of performing western blotting under GLUT4 depletion.

- Fig. 1C, right: I guess that the legend of the ordinate should be total GLUT4 instead of pmGLUT4?

Response: This has been rectified.

- Fig. 2B: the legend is unclear. What are the three tracks for each condition?

Response: This has been rectified.

The figure legend:

(B) Total cellular GLUT4 was assessed by Western Blot (upper panel). Each track represents an independent biological replicate, from left to right control, Dexamethasone (Dexa), Tumour Necrosis Factor (TNF) and Chronic insulin (CI). Quantitation (lower panel) of GLUT4 normalised to control cells (set to 100%). Data are mean \pm S.D., n = 3 with * p <0.05 compared to control by 1way ANOVA with Dunnet's test for multiple comparisons.

- Discussion (1st page): Fig. 8 does not exist.

Response: This has been rectified - changed to Figure 7.

September 23, 2022

RE: Life Science Alliance Manuscript #LSA-2022-01585-TR

Dr. James G Burchfield
University of Sydney
The Charles Perkins Centre
The University of Sydney
Sydney, NSW 2006
Australia

Dear Dr. Burchfield,

Thank you for submitting your revised manuscript entitled "A high-content endogenous GLUT4 trafficking assay reveals new aspects of adipocyte biology". We would be happy to publish your paper in Life Science Alliance pending final revisions necessary to meet our formatting guidelines.

- please upload your main manuscript text as an editable doc file
- please upload both your main and your supplementary figures as single files
- please note that LSA allows two corresponding authors-one corresponding author and one secondary corresponding author- and both corresponding authors must add their ORCID ID; you should have received instructions on how to do so
- please add the Twitter handle of your host institute/organization as well as your own or/and one of the authors in our system
- the Right Retention Statement is unnecessary and should be removed. The grant mention can go under Acknowledgements. All content in Life Science Alliance is published as open access with a CC-BY license. Copyright is retained by the authors.
- we encourage you to introduce your panels in your figure legends in alphabetical order
- please add a callout for Figure 2I; Figure 7B; Figure S3B,C,E; Figure S5E to your main manuscript text
- please check your figure callouts on page 12 and make sure each callout also includes a figure number; you have a callout that reads (Fig. SD, E)

Figure Check:

- Figure S3 figure Legend is missing panel B
- Figure 5D needs a scale bar

A. FINAL FILES:

B. MANUSCRIPT ORGANIZATION AND FORMATTING:

Sincerely,

October 10, 2022

RE: Life Science Alliance Manuscript #LSA-2022-01585-TRR

Dr. James G Burchfield
University of Sydney
The Charles Perkins Centre
The University of Sydney
Sydney, NSW 2006
Australia

Dear Dr. Burchfield,

Thank you for submitting your Methods entitled "A high-content endogenous GLUT4 trafficking assay reveals new aspects of adipocyte biology". It is a pleasure to let you know that your manuscript is now accepted for publication in Life Science Alliance. Congratulations on this interesting work.

DISTRIBUTION OF MATERIALS:

Again, congratulations on a very nice paper. I hope you found the review process to be constructive and are pleased with how the manuscript was handled editorially. We look forward to future exciting submissions from your lab.

Sincerely,
